# Circadian clock regulator Bmal1 gates axon regeneration via Tet3 epigenetics in mouse sensory neurons

Dalia Halawani [1,5], Yiqun Wang [1,2,5], Aarthi Ramakrishnan[1], Molly Estill[1], Xijing He [2,3], Li Shen [1], Roland H. Friedel [1,4] & Hongyan Zou [1,4] ✉

Axon regeneration of dorsal root ganglia (DRG) neurons after peripheral axotomy involves reconfiguration of gene regulatory circuits to establish regenerative gene programs. However, the underlying mechanisms remain unclear. Here, through an unbiased survey, we show that the binding motif of Bmal1, a central transcription factor of the circadian clock, is enriched in differentially hydroxymethylated regions (DhMRs) of mouse DRG after peripheral lesion. By applying conditional deletion of *Bmal1* in neurons, in vitro and in vivo neurite outgrowth assays, as well as transcriptomic profiling, we demonstrate that Bmal1 inhibits axon regeneration, in part through a functional link with the epigenetic factor Tet3. Mechanistically, we reveal that Bmal1 acts as a gatekeeper of neuroepigenetic responses to axonal injury by limiting Tet3 expression and restricting 5hmC modifications. Bmal1-regulated genes not only concern axon growth, but also stress responses and energy homeostasis. Furthermore, we uncover an epigenetic rhythm of diurnal oscillation of Tet3 and 5hmC levels in DRG neurons, corresponding to time-of-day effect on axon growth potential. Collectively, our studies demonstrate that targeting Bmal1 enhances axon regeneration.

Unlike injury in the central nervous system (CNS), peripheral nerve injury triggers axon regeneration, but the repair process is often slow and incomplete. A crucial step for successful nerve repair is to accelerate axon regeneration[1], but the gene regulatory mechanisms that enable injured neurons to rapidly switch into a growth state remain poorly defined[2]. Emerging evidence also implicates neuroimmune interactions in augmenting axon regeneration;[3–6] how neuronal intrinsic and extrinsic injury and stress responses are orchestrated is unclear. Such knowledge may help advance new strategies to enhance nerve repair, with translational relevance for CNS injury.

Dorsal root ganglia (DRG) sensory neurons represent a preferred system to study axon regeneration, as their central and peripheral axon branches respond differently to axotomy. While central lesion

(CL) after spinal cord injury leaves DRG neurons in a state that is refractory to axon regeneration, peripheral lesion (PL) triggers a regenerative response that enables regeneration of not only peripheral but also central branches[7,8]. This so-called "conditioning lesion" effect is transcription dependent[9], thus presenting a starting point to investigate regulatory mechanisms of regeneration-associated genes (RAGs)[2,10,11].

Works from our laboratory and others have demonstrated that conditioning lesion initiates a genome-wide reconfiguration of histone modifications and DNA methylation, thus adding a layer of epigenetic control for robust and fine-tuned gene regulation[2,10,12–15]. Tet methylcytosine dioxygenase 3 (Tet3) has emerged as a critical regulator of axon regeneration in conditioned DRG neurons. Tet3 catalyzes the

[1]Nash Family Department of Neuroscience, Friedman Brain Institute, Icahn School of Medicine at Mount Sinai, New York, NY, USA. [2]Department of Orthopedics, Second Affiliated Hospital of Xi'an Jiaotong University, Shaanxi, China. [3]Department of Orthopedics, Xi'an International Medical Center Hospital, Xi'an, China. [4]Department of Neurosurgery, Icahn School of Medicine at Mount Sinai, New York, NY, USA. [5]These authors contributed equally: Dalia Halawani, Yiqun Wang. ✉e-mail: hongyan.zou@mssm.edu

iterative oxidation of methylcytosine (5mC) to 5-hydroxymethylcytosine (5hmC), formylcytosine (5fC), and carboxylcytosine (5caC), which is then converted back to 5C by base excision repair (BER)[16]. We previously showed that PL induces an upregulation of Tet3, but not Tet1 or Tet2, in axotomized DRG neurons along with global 5hmC gains[17]. DNA demethylation is required for the conditioning effect, as Tet3 silencing reduces global 5hmC levels and diminishes axon regrowth capacity[18]. Aside from being an intermediate of the DNA demethylation cascade, 5hmC is also a stable DNA base with unique gene regulatory functions in its own right. Notably, 5hmC is highly abundant in the CNS, accounting for ~40% of modified cytosines, mostly in gene body regions[19–22], and is pivotal for neuronal differentiation[22,23]. While these earlier studies showed the necessity of Tet3 and DNA demethylation for axon regeneration, the regulators of Tet3 induction and how Tet3 selects the target sites for DNA demethylation remain unresolved. It is also unclear if Tet3 augmentation and the resultant 5hmC gain can promote regenerative capacity.

Here, in search for transcription factors (TFs) that may affect Tet3-5hmC responsiveness to axonal injury, we performed an unbiased screen for over-represented TF binding motifs within the differentially hydroxymethylated regions (DhMRs) of conditioned DRG. Intriguingly, we identified an enrichment of the binding motif for Bmal1, a central regulator of the circadian clock[24]. By applying conditional gene deletion in conjunction with in vitro and in vivo models of axon regrowth, we demonstrated that neuronal knockout of *Bmal1* enhanced axon regeneration partly through a Tet3-dependent mechanism. Transcriptomics profiling revealed that the Bmal1 regulon after peripheral axotomy not only concerned axon regrowth, but also stress responses, energy homeostasis, and neuroinflammation. Interestingly, Tet3 and 5hmC displayed diurnal oscillations in DRG neurons, corresponding to a time-of-day effect on axon growth potential. Our study thus raises awareness of the time-of-day effect on neural regenerative responses, with ramifications for experimental designs and translational strategies for neural repair.

## Results

### Bmal1 binding motif is enriched in DhMRs of conditioned DRG

To explore the functional relevance of DNA hydroxymethylation in axon regeneration, we previously performed genome-wide mapping of 5hmC modifications in axotomized DRG, which identified 1,036 DhMRs specific for peripheral lesion (PL, regenerative state) as compared to dorsal column lesion (DCL, non-regenerative state) or no injury[17] (Fig. 1a). Among the PL-DhMRs (each ~200 bp), 561 (54%) displayed 5hmC gain and 475 (46%) loss, both predominantly localized in gene body and intergenic regions rather than promoters and CpG islands (Fig. 1b), suggesting a complex role in gene regulation.

To identify transcription factors (TFs) that may collaborate with Tet3 for 5hmC modifications or display methylation-sensitive DNA binding, we searched for statistically significant over-represented TF binding motifs in PL-DhMRs, and found that the top enriched were the binding motifs for bHLH-PAS, a TF family functioning as molecular sensors of physiological and environmental stimuli, including circadian rhythm, hypoxia, and xenometabolites[25,26] (Fig. 1c). Specifically, 39% of PL-DhMRs harbored binding motifs for TFs associated with circadian clock, including Bmal1, Clock, Npas2 (a Clock homolog), as well as Bhlhe40 and 41, while 42% contained hypoxia-response-elements (HRE) accessible by hypoxia-inducible factors, i.e., HIF1α, 2α, and ARNT (HIF1β) (Figs. 1c, d, S1a; Supplementary Data 1). Other enriched TF binding motifs included Notch (RBPJ)[27], members of bHLH TF family (e.g., MYC, USF1/2, BCL11a), as well as thyroid hormone receptor (THR), which is known to interact with Tet3[28] (Fig. 1c).

We next analyzed the PL-DhMRs with 5hmC gain or loss separately, which identified shared (e.g., ARNT, NPAS1) but also distinct enriched bHLH-PAS binding motifs (Fig. S1b). Notably, many of these bHLH-PAS TFs belonged to the RAGs induced by the conditioning

lesion, implicating a role in axon regeneration (Fig. S1c). Indeed, activation of HIF1α (a bHLH-PAS TF) through intermittent hypoxia promotes axon regrowth of DRG neurons[29].

### Conditioning lesion enhances chromatin accessibility at DhMRs

Since DNA methylation alters chromatin landscape[30], we next examined a recent ATAC-seq dataset on DRG before and after PL[14]. This revealed that PL led to increased chromatin accessibility at DhMRs (more prominent for the ones with 5hmC gain) and transcription start site (TSS) of the associated genes (Fig. S1d). Our previous work has shown that genes harboring PL-DhMRs are enriched for pathways linked to axon regeneration (e.g., CREB and Neurotrophin)[17], here we analyzed the genes displaying 5hmC gain or loss separately, which showed that the genes in the 5hmC gain category (n = 988) concerned epithelial cell proliferation and differentiation, organ regeneration, and lymphocyte co-stimulation, while the genes in the 5hmC loss category (n = 834) mainly concerned metabolism (e.g., response to thyroid hormone, amino acid biosynthetic, glucose homeostasis) and cytoskeletal reorganization (Fig. S1d; Supplementary Data 2). In this context, it is noteworthy that as an intermediate of the DNA demethylation cascade, 5hmC gain or loss does not strictly predict transcriptional gain or loss[17].

Tet3 has two isoforms due to alternative transcriptional start sites[31–33]. We thus examined their relative abundance in DRG, which showed that the short form (Tet3-S), which lacks the CXXC DNA-binding domain, is the predominant form (Fig. 1e). Hence, Tet3 recruitment to target genomic loci for DNA hydroxymethylation upon peripheral axotomy might involve cooperation with TFs.

We next focused on Bmal1, an essential and the only non-redundant core regulator of the mammalian circadian clock;[34] its function in axon regeneration is unknown. We first examined the PL-DhMRs containing Bmal1 binding motifs (n = 341) and the associated genes (n = 616, identified by GREAT[35]) (Fig. 1f, g, Supplementary Data 3). Pathway enrichment analysis showed that these genes featured well-known pro-regenerative pathways, e.g., IGF-1, BMP, NGF, Axon guidance, and PTEN[36,37] (Fig. 1g). Bmal1 is known as a pioneer-like TF (i.e., modulating chromatin structure to recruit other TFs)[38,39], we thus re-examined the DRG ATAC-seq dataset, which indeed revealed that PL resulted in increased chromatin accessibility at these DhMRs and the TSS of the associated genes (Fig. 1h), consistent with epigenetic changes in regenerating DRG (Fig. 1f).

We wondered whether there might be a potential protein-protein interaction between Tet3 and Bmal1. Co-immunoprecipitation (co-IP) with a specific antibody against Tet3[40–42] demonstrated pull down of Tet3 along with Bmal1 in Neuro-2a cells and in DRG lysates (Fig. S2a, b). Unlike in Neuro-2a lysate with a predominant band of Tet3 at ~185 kDa detected in Western blotting (WB) (Fig. S2c), in whole DRG lysates WB showed a dominant band of Tet3 at ~80 kDa, while Tet3 pull down resulted in a faint band at ~185 kDa and multiple bands at lower molecular weights (Fig. S2b), which may reflect proteolytic processing of Tet3[43]. Attempts to reverse co-IP with a specific antibody against C-terminal region of Bmal1 failed to pull down either endogenous or overexpressed Tet3 in Neuro-2a cells (Fig. S2c), possibly related to antigen masking from Tet3 binding to the C-terminal domain of Bmal1 which contains a conserved transactivation domain involved in binding to transcriptional regulators[44,45].

### Conditioning lesion downregulates Bmal1 in DRG neurons

To investigate the relevance of Bmal1 in axon regeneration, we first examined how axonal injury might affect the expression of interconnected clock genes of the translational/transcriptional feedback loops that control Bmal1 expression and transcriptional activity (Fig. 2a). To this end, we performed RNA-sequencing (RNA-seq) to profile differentially expressed genes (DEGs) in DRG in response to PL (Supplementary Data 4). We found that *Bmal1* (a.k.a. *Arntl*) was

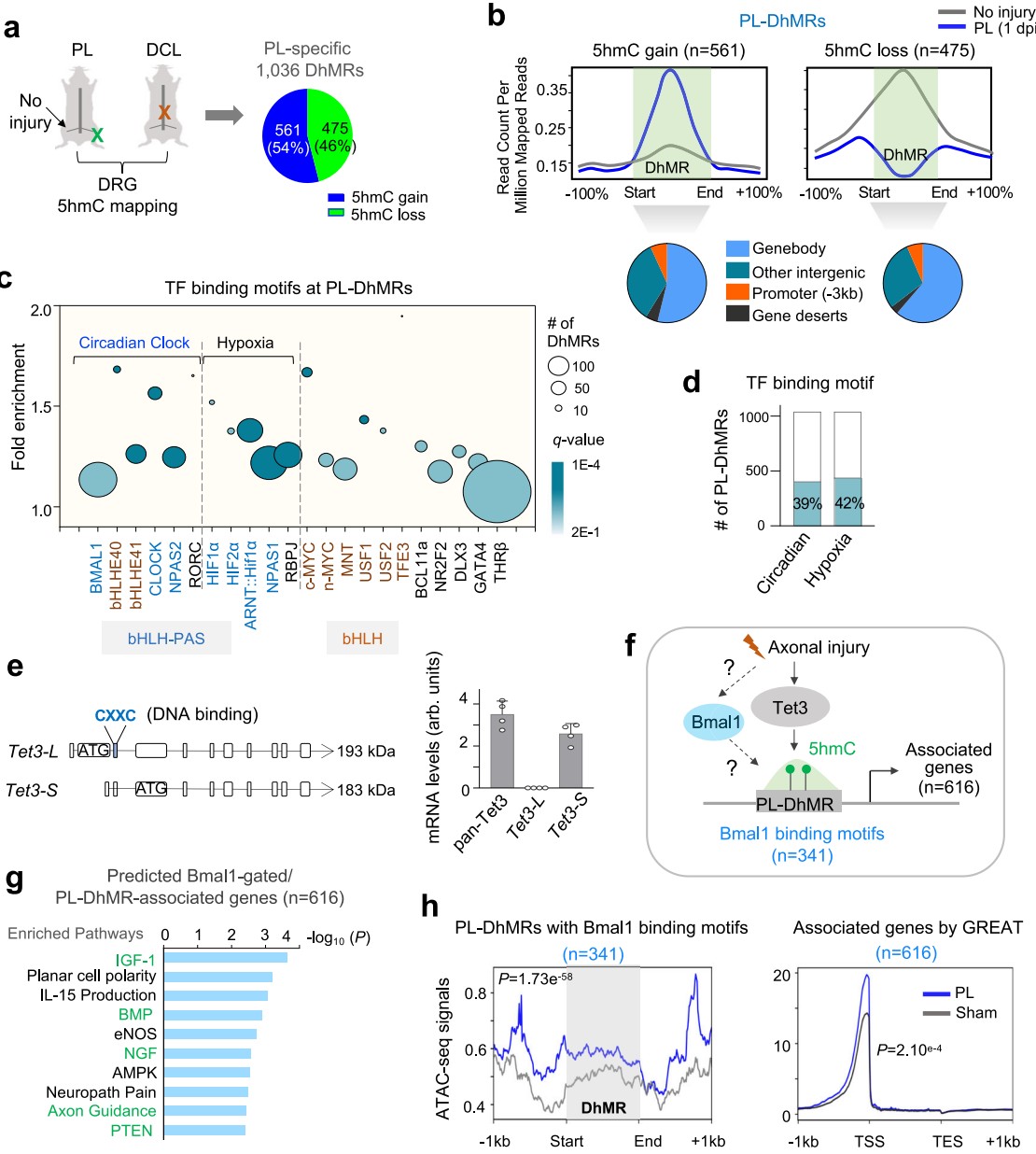

**Fig. 1 | Bmal1 is linked to peripheral lesion of DRG and interacts with Tet3.**
**a** Graphical summary of genome-wide mapping of 5hmC reconfigurations after PL (peripheral lesion) vs. DCL (dorsal column lesion) or no injury, which identified 1,036 differentially hydroxymethylated regions (DhMRs) after PL[17]. Pie chart shows proportion of PL-DhMRs with 5hmC gain versus loss. **b** Top, normalized 5hmC signals after PL versus no injury across DhMRs[17]. Bottom, pie charts showing distribution of DhMRs in genomic regions. **c** Bubble plot showing enriched TF binding motifs in PL-specific DhMRs. Members of the bHLH-PAS family are labeled in blue and members of the bHLH family in brown. Circle size indicates DhMR numbers and color scale indicates significance of enrichment. **d** Bar graphs showing the number (y-axis) and percentage of PL-specific DhMRs that contain binding motifs for circadian-related or hypoxia-related transcription factors. **e** Left, diagram of the

long and short isoforms of *Tet3*. ATG denotes start codon, CXXC indicates DNA binding domain. Right, qRT-PCR shows predominant expression of short isoform Tet3-S in uninjured DRGs. Data represent mean ± SEM. *n* = 4 independent studies, each with 6-9 pooled DRGs. **f** Model of potential Bmal1 and Tet3 interaction at DhMRs that may impact Tet3 targeting and 5hmC reconfigurations after peripheral lesion of DRG. **g** Ingenuity pathway analysis (IPA) highlighting the top enriched pathways for the 616 genes harboring PL-specific DhMRs with Bmal1 binding motif in response to axonal injury. Right-tailed Fisher's exact test. **h** Analysis of ATAC-seq signals from DRG after sham or PL at 1 dpi (raw data from Palmisano et al.[14]) showed increased chromatin accessibility at PL-specific DhMRs with Bmal1 binding motif and associated genes near transcription start sites (TSS). TES, transcription end site. Welch's two-sample *t*-test.

significantly downregulated at 1 day post-injury (1 dpi) relative to no injury (Fig. 2b), in agreement with a published DRG RNA-seq dataset[14]. The result was corroborated by qRT-PCR analysis (Fig. 2c). Surveying a recent single cell RNA-seq dataset[46] revealed that 8 out of the 9 sub-clusters of DRG neurons downregulated *Bmal1* at 24 h post-PL (Fig. S3a).

Bmal1 (the β subunit) heterodimerizes with Clock (the α subunit) to form a transcriptional complex that binds to Enhancer (E)-box

motifs to regulate a large repertoire of clock-controlled genes (CCGs) as well as its own feedback regulators (Fig. 2a). These include CRY and PER, both negative regulators of Bmal1 transcriptional activity[47], as well as REV-ERBα/β, two orphan nuclear receptors constituting a second feedback loop, exerting negative effect on *Bmal1* transcription[48]. We found that *Per1 and Cry2*, but not *Cry1*, were upregulated after PL (Fig. 2b, c). Time course analysis showed that *Bmal1* expression was downregulated at 12 h post-injury, but returned

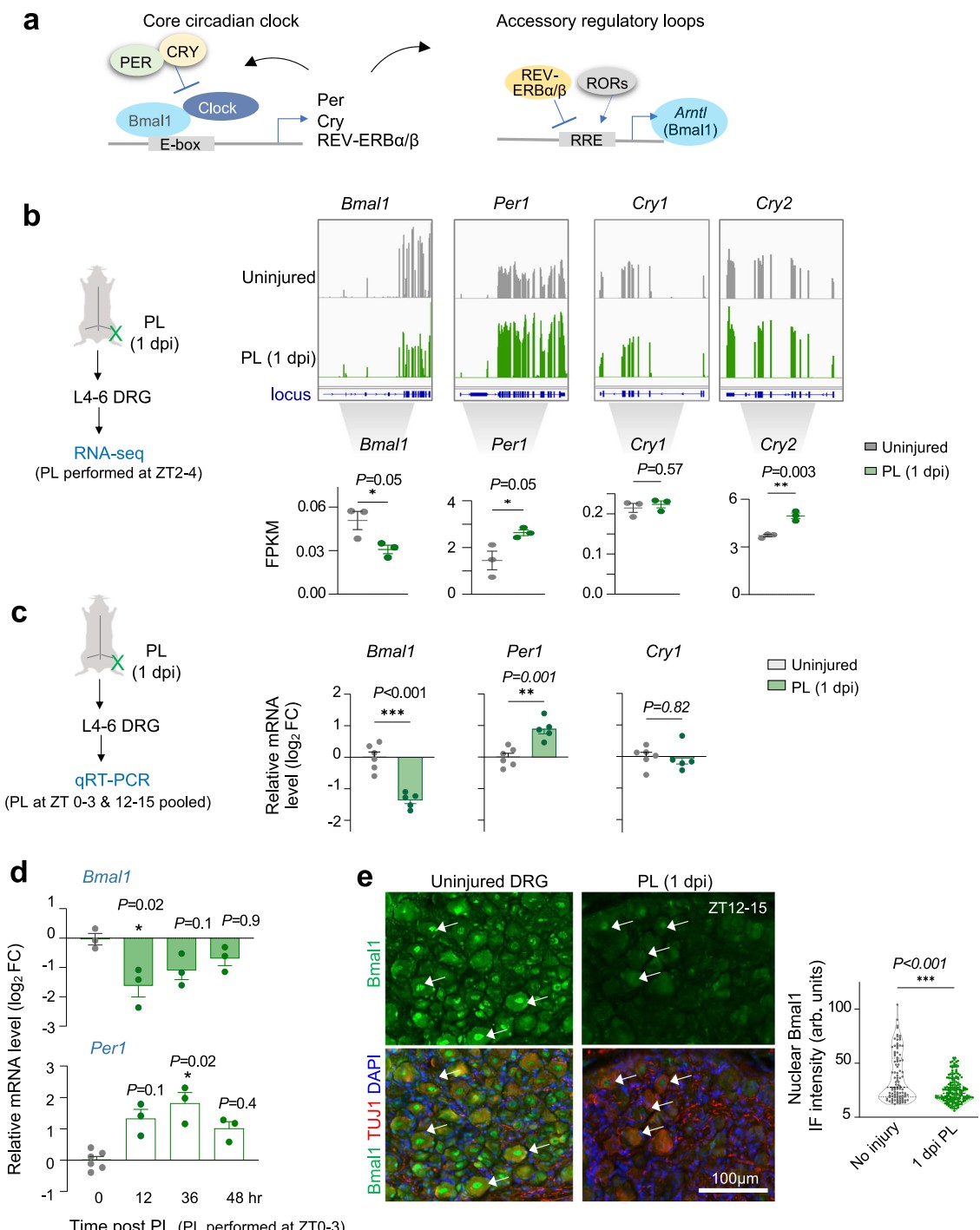

**Fig. 2 | Peripheral lesion of DRG axons results in downregulation of Bmal1.**
**a** Schematic of core circadian clock components and feedback loops that control Bmal1 expression and activity. E-box, enhancer box; RRE, REV-ERB/ROR response element. **b** Left, experimental paradigm of ipsilateral L4–L6 DRGs analyzed by RNA-seq at 1-day after peripheral lesion (PL at ZT2-4) or no injury. Top right, RNA-seq coverage tracks for differentially expressed circadian genes after PL at 1 dpi pooled from $n = 3$ samples per group. Bottom right, the expression levels of circadian clock genes as fragments-per-kilobase-per-million reads (FPKM). $n = 3$, each pooled from L4–L6 DRGs. Data represent mean ± SEM. Unpaired two-tailed Student's $t$-test. **c** Left, experimental paradigm of peripheral lesion (PL) and collection of ipsilateral L4–L6 DRGs for gene expression analysis. Right, qRT-PCR shows transcriptional

changes of circadian clock genes in axotomized DRG at 1 dpi after PL as compared to uninjured DRG. $n = 5$–6 independent samples, each pooled from L4–L6 DRGs. Data represent mean ± SEM. Unpaired two-tailed Student's $t$-test. **d** Time-course analysis of *Bmal1* transcription in L4–6 DRGs after PL by qRT-PCR. $n = 3$ independent DRG samples, pooled from two CD1 mice. Data represent mean ± SEM. Kruskal–Wallis with Dunn's multiple comparison's test. **e** Representative IF images and quantification of L5 DRGs show diminished nuclear signal of Bmal1 in axotomized DRG neurons (TUJ1$^+$) at 1 dpi after PL. Violin plots of $n = 100$ uninjured and 183 axotomized neurons from 4 DRGs from 4 mice per group. Median is shown by black line. Mann–Whitney test.

to baseline level by 36 h, while *Per1* was upregulated at both time points (Fig. 2d).

We next conducted immunofluorescence (IF) staining, which confirmed diminished Bmal1 nuclear immunosignals in conditioned DRG neurons at 1 dpi as compared to no injury (Fig. 2e). Of note, WB did not reveal a significant reduction of Bmal1 protein levels in DRG tissue lysates at 12 or 24 h post-injury (Fig. S3b), possibly due to a high abundance of glial cells (neurons representing only 12.5% of total cells in DRG[6]). IF for Tet3 and 5hmC showed that both were increased in axotomized DRG neurons at 1 dpi (Fig. S3c–e), in agreement with previous works from the Ming laboratory and our own[17,18]. Concordantly, *Tet3* mRNA levels were higher after PL, and scRNA-seq confirmed *Tet3* induction in all DRG neuronal subtypes (Fig. S3a, c). Hence, conditioning lesion of DRG affects the expression of molecular oscillators, with Bmal1 activity transiently suppressed as evidenced by its reduced nuclear levels and induction of *Per1* and *Cry2*, both negative regulators of Bmal1 transcriptional activity[47].

## Bmal1 inhibition enhances neurite outgrowth

To understand the functional significance of *Bmal1* downregulation by the conditioning lesion, we first tested the effect of silencing *Bmal1* in Neuro-2a cells (Fig. S4a), which led to enhanced neurite outgrowth (Fig. 3a), whereas forced over-expression of Bmal1 reduced neurite elongation but not neurite initiation, evidenced by comparable proportions of axon-bearing cells (Figs. 3b, S4b). Likewise, in induced neurons derived from human embryonic stem (hES) cells wherein the clock was operational[49], pharmacological stabilization of PER and CRY (both negative regulators of Bmal1)[50,51] also enhanced neurite outgrowth (Figs. 3c, S4c, d). These culture results support an inhibitory role of Bmal1 for axon growth.

To study the function of Bmal1 in vivo, we generated *Bmal1^fl/fl^ Thy1-CreER^T2^* mice for neuron-specific, tamoxifen-inducible conditional knockout (cKO) of *Bmal1*. This strategy circumvents developmental phenotypes of *Bmal1* deletion with the Nestin-Cre line, which results in neurodegeneration[52]. As the *Thy1-CRE^ERT2^/EYFP* (SLICK-H) transgenic line drives pan-neuronal expression of both CreER^T2^ recombinase and EYFP[53,54], we first confirmed uniform expression of EYFP in DRG neurons with fluorescence imaging of DRG tissue sections and primary DRG neuron cultures (Fig. S5a, b).

We then administered tamoxifen to adult (8–12 weeks) *Bmal1^cKO^* mice and littermate controls (*Bmal1^fl/fl^* or *Bmal1^fl/+^*) and analyzed DRGs two weeks later. Genotyping of DRGs confirmed excision of the floxed region of the *Bmal1* gene, with *Bmal1^fl/fl^ Nestin-Cre* (*Bmal1^Nes-Cre^*) mice serving as positive control[52] (Fig. S5c). qRT-PCR verified a significant downregulation of *Bmal1* and its target genes *Nr1d1* and *Nr1d2* (encoding REV-ERBα and β, respectively) in *Bmal1^cKO^* DRGs relative to controls, while *Fabp7*, a target gene of REV-ERB transcriptional repression[55–57], was upregulated (Fig. S5d). The transcriptional changes of REV-ERBs as a readout of *Bmal1* ablation agreed with previous studies using the Nestin-Cre *Bmal1* cKO[52,55]. Other clock components such as *Clock*, *Per1*, *Per2*, and *Cry1* did not show significant transcriptional changes in DRG tissues with neuron-specific *Bmal1* cKO (Fig. S5d). IF staining further confirmed reduced Bmal1 immunosignals in DRG neurons in both *Bmal1^Thy1-CreERT2^* and *Bmal1^Nes-Cre^* cKO mice (Fig. S5e, f).

Next, we conducted neurite outgrowth assays with primary DRG neurons, which gauge axon regrowth potential, as the dissociation process results in axotomy (Fig. 3d). Aligned with the above results, DRG neurons from *Bmal1^cKO^* mice extended longer neurites than controls at 22 h in vitro (mean $311 \pm 10$ vs. $260 \pm 7\,\mu m$) (Fig. 3e). Similar results were also obtained with primary DRG neurons from *Bmal1^Nes-Cre^* mice (Fig. S6a). The proportion of neurite-bearing neurons was not different between genotypes, signifying comparable axon initiation capacity (Fig. S6b).

The neurite length of *Bmal1*-deficient DRG neurons did not reach that of conditioned DRG neurons with prior PL (Fig. 3e). This indicated

that the conditioning effect also operates through Bmal1-independent mechanisms. However, *Bmal1* deletion could further augment the conditioning effect, evidenced by even longer axons of conditioned *Bmal1^cKO^* DRG neurons than conditioned controls at 10 h post-seeding (mean $329 \pm 144$ vs. $278 \pm 121\,\mu m$) (Fig. 3f). On the other hand, Bmal1 over-expression in primary DRG neurons did not cause significant reduction of neurite outgrowth, unlike in Neuro-2a cells (Fig. S6c, and see Fig. 3b), suggesting that the conditioning effect can bypass or overcome the Bmal1 block. The growth promoting effects of *Bmal1* deletion were also confirmed in adult DRG explant cultures, which allow analysis after a longer growth period of 4 days after seeding (Fig. 3g).

We next compared in vivo axon regeneration in a sciatic nerve crush injury model (Fig. 4a). IF staining for SCG10, a well-established axon regeneration marker that specifically labels regenerating but not uninjured sensory axons[14,18,29,58], showed that *Bmal1^cKO^* mice displayed accelerated axonal regrowth at 1 dpi and 3 dpi, with ~50% longer SCG10+ axons than in control (*Bmal1^fl/fl^* or *Bmal1^fl/+^*) that received identical tamoxifen injection (Fig. 4b–e). Regeneration index calculation confirmed the enhanced axon regenerative phenotype of *Bmal1^cKO^* animals (Fig. 4c, e). These results were corroborated in *Bmal1^Nes-Cre^* cKO mice (Fig. S6d, e). We also examined re-innervation of the epidermal layer of the glabrous skin of the foot pad after sciatic nerve injury (Fig. 4f). By 21 dpi, whole mount IF for the neuronal marker PGP9.5 revealed that cutaneous reinnervation area reached over twice as large in *Bmal1^cKO^* mice as in controls (Fig. 4f, g). This also held true for regenerated axon branches labeled by neurofilament-H (NF-H)+ (Fig. 4h), signifying a higher abundance of maturing axons in the reinnervated epidermis[59]. Together, these results supported an inhibitory role of Bmal1 in axon regeneration and that the conditioning lesion operates partly by suppressing the Bmal1 blockade (Fig. 4i).

## Bmal1 ablation augments Tet3 and 5hmC induction in conditioned DRG neurons

To pinpoint the mechanisms by which Bmal1 restricts axon regeneration, we tested the hypothesis that Bmal1 gates 5hmC responsiveness to axonal injury. We first examined uninjured DRG neurons and observed higher baseline levels of Tet3 and 5hmC in Bmal1-deficient DRG neurons relative to controls (Fig. S7a, b). Upon PL, their upregulation appeared more pronounced in *Bmal1^cKO^* DRG neurons than controls at 3 dpi (Fig. 5a, b), a finding also observed in primary DRG neurons (axotomized during dissociation) (Fig. S7c). The phenotypes were sustained at 21 dpi after PL, signifying long-lasting epigenetic changes with *Bmal1* deletion (Fig. S7d, e).

Given the global 5hmC elevation in Bmal1-deficient DRG neurons, we initially anticipated induction of a wide range of RAGs, but surveying RAGs in uninjured DRGs from *Bmal1^cKO^* vs. control mice revealed no significant upregulation of well-known RAGs such as *Atf3* (a marker of conditioning lesion[60]), neurotrophins (*Bdnf*, *Ngf*, *Ntf3*), *Gal*, or *Gap43* (Fig. 5c). We thus posited that *Bmal1* deletion and the concomitant Tet3-5hmC epigenetic changes may prime DRGs for regenerative responses to axonal injury. Indeed, in axotomized DRGs at 1 day after PL, we detected increased transcription of *Bdnf*, *Gal*, and *Npy* in *Bmal1^cKO^* DRGs relative to controls (Fig. 5c), while IF staining verified higher protein level of NPY in mutant DRG neurons at 3 dpi (Fig. 5d). In line with the IF results shown in Fig. 5a, *Tet3* transcription was higher in *Bmal1^cKO^* DRG (Fig. 5c). Of note, not all RAGs were affected, e.g., GAP43 or ATF3 immunosignals appeared comparable between genotypes at baseline or 3 dpi (Fig. S8a–c).

To assess if Tet3 is required for the axon promoting effect of *Bmal1* deletion, we first took advantage of C35, a newly characterized cell-permeable inhibitor of Tet catalytic activity[61]. Exposure to C35 reduced 5hmC levels in DRG neurons (Fig. S8d) and attenuated axon elongation phenotype of *Bmal1^cKO^* neurons in a dose-dependent manner (Fig. S8e, f). We next tested *Tet3* knockdown by siRNA

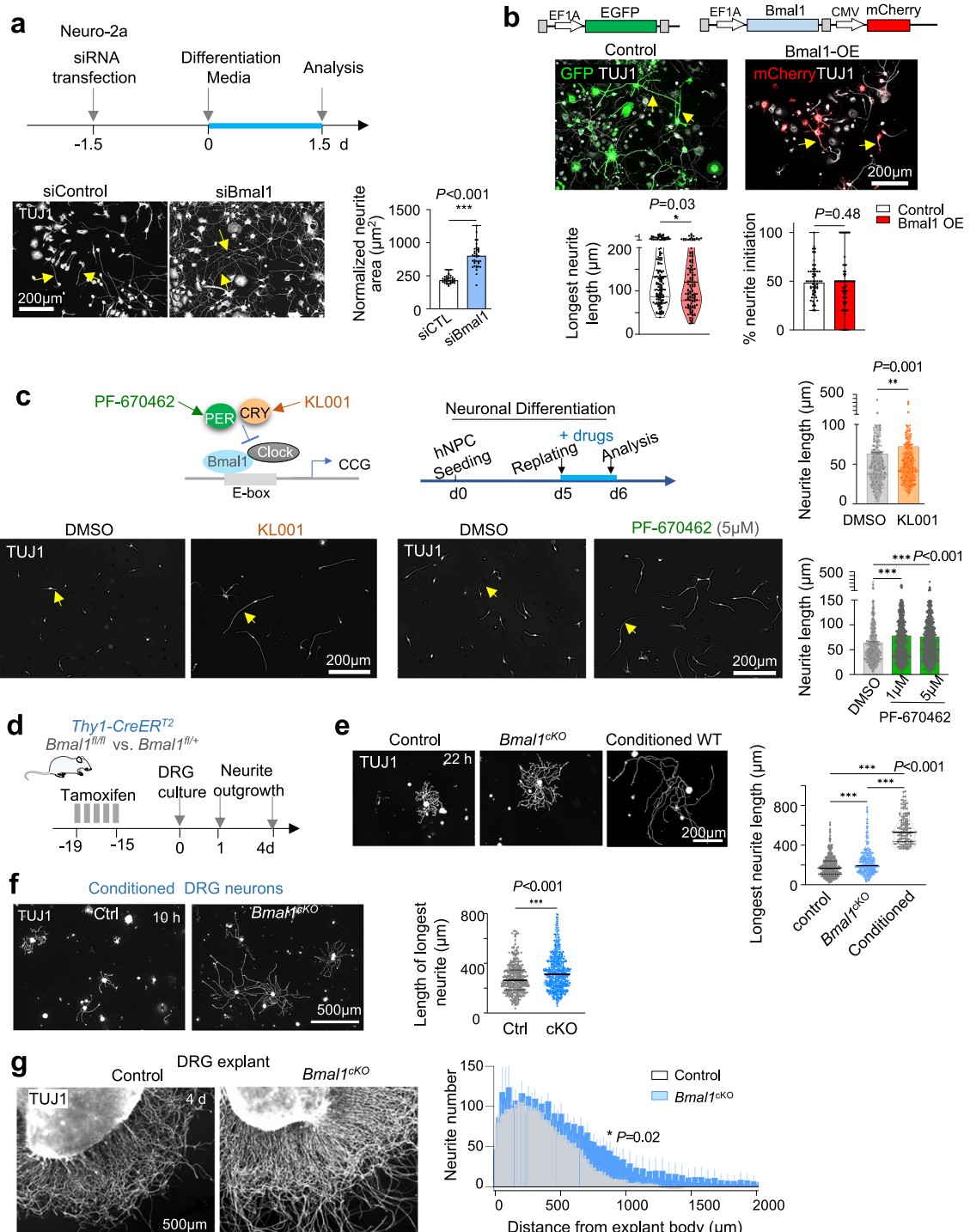

**Fig. 3 | Bmal1 inhibition enhances axon outgrowth. a** Top, experimental paradigm. Bottom, representative IF images and quantification of TUJ1⁺ area normalized to the number of soma per image, $n = 39$ images from 2 independent experiment. Data represent mean ± SEM, Mann–Whitney test. **b** Top, schematic of lentiviral vectors. Bottom, representative IF images and quantification of neurite length and percentage of neurite initiation of control (EGFP⁺) or Bmal1 overexpression (mCherry⁺) in Neuro-2a cells. n > 140 neurons for neurite length, and 60 images per group for percentage of neurite initiation. Data represent mean ± SEM. Mann–Whitney test. **c** Top left, diagram of pharmacological drugs stabilizing negative regulators of Bmal1 transcriptional activity. Top right, paradigm of neurite outgrowth assay of iNeurons. Bottom, representative IF images show longer neurite outgrowth from drug-treated iNeurons. Arrows point to neurites. Quantification: $n = 600$–900 neurons pooled from 3 independent cultures. Data represent mean ± SEM. Mann–Whitney test for KL001 vs. DMSO. Kruskal–Wallis with Dunn's multiple comparison test for PF-670462 vs. DMSO. **d** Experimental

paradigm for tamoxifen administration to adult mice for neuron specific *Bmal1* cKO. **e** IF images and quantification of neurite outgrowth of primary DRG neurons. Violin plots of $n = 348$ control, 305 *Bmal1$^{cKO}$*, and 148 conditioned neurons from $n = 3$ mice. Median is shown by black line. Kruskal–Wallis with Dunn's multiple comparison test. **f** IF images show that *Bmal1* cKO could augment DRG conditioning lesion effect (with PL 3 days prior) in enhancing neurite outgrowth. Note early time point at 10 h after plating due to exuberant neurite outgrowth from the conditioning effect. Violin plots of $n = 375$ control and 626 *Bmal1$^{cKO}$* neurons collected from $n = 3$ mice per group. Data represent mean ± SEM. Two-way ANOVA followed by Bonferroni's multiple comparison test. **g** IF images of adult DRG explants show longer neurite outgrowth from *Bmal1$^{cKO}$* DRGs compared to control at 4 d of culture. Quantification of 8 images collected from $n = 6$ DRGs (L4–L6) from $n = 2$ mice per group. Two-way ANOVA followed by Bonferroni's multiple comparison test.

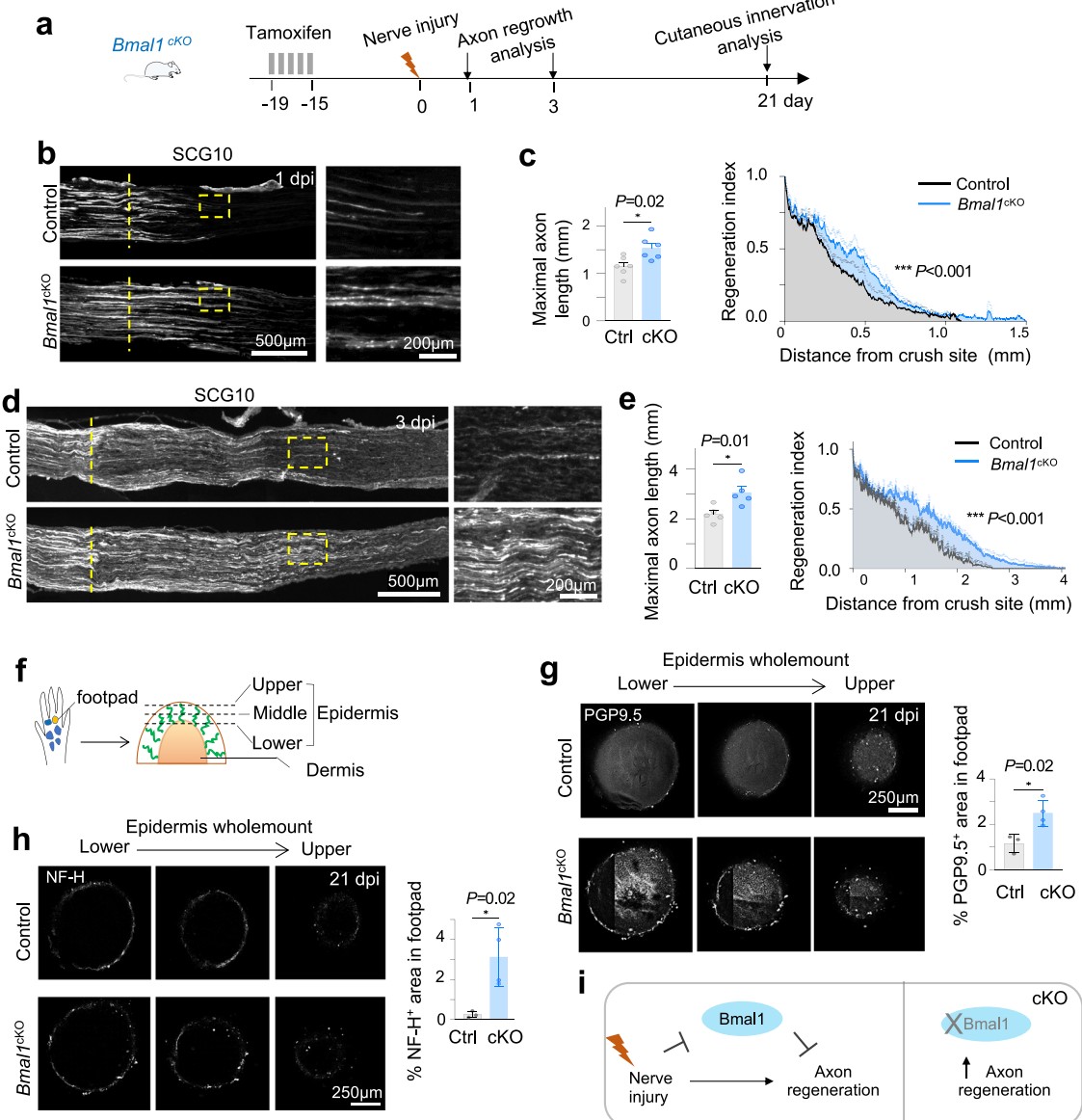

**Fig. 4 | Neuronal deletion of *Bmal1* accelerates axon regeneration after sciatic nerve injury. a** Experimental paradigm of in vivo axon regeneration analysis at 1, 3, and 21 d after sciatic nerve injury. Representative IF images and quantifications of axon regeneration (SCG10⁺) at sciatic nerve crush site at 1 dpi (**b**, **c**) or 3 dpi (**d**, **e**). Note enhanced axon regeneration in *Bmal1* cKO as compared to littermate controls. Dashed vertical lines denote lesion center defined by maximal SCG10 immunointensity. Right panels show enlarged images of boxed areas. Data represent mean ± SEM and unpaired two-tailed Student's *t* test for maximal axon length. Two-way ANOVA for regeneration index, measured by SCG10 immunointensity

normalized to value at lesion center. *n* = 5–6 mice per genotype. **f** Diagram of analysis of innervation of footpad epidermis. Representative images and quantifications of whole mount IF staining for pan-neuronal marker PGP9.5 (**g**) or axon marker NF-H (**h**) show improved footpad epidermis innervation in *Bmal1* cKO at 21 dpi. Micrographs are representative confocal scans of a 400 μm segment, spanning the top, middle, and lower portion of the epidermis. *n* = 3 control and 4 *Bmal1^cKO* mice per group. Data represent mean ± SEM. Unpaired two-tailed Student's *t*-test. **i** Working model for a role of Bmal1 as repressor of axon regeneration; removing this brake augments axon regeneration after peripheral nerve injury.

(Fig. S8g), which similarly reduced 5hmC levels and the growth advantage of *Bmal1^cKO* DRG neurons (Figs. 5e, f, S8h). Hence, the promoting effect of *Bmal1* deletion requires Tet3.

Since *Bmal1* ablation can increase reactive oxygen species (ROS)[52], which may restrict $Fe^{2+}$ availability required for Tet3 catalytic activity[62–64], we tested the effect of N-acetylcysteine amide (NACA), a ROS inhibitor with free radical scavenging property[65]. We found that while NACA treatment led to a modest increase of 5hmC in control DRG neurons, the effect was more pronounced in *Bmal1^cKO* neurons (Fig. 5g). This suggested that even though Tet3 was upregulated in *Bmal1^cKO* DRG neurons, there appears to be a built-in safeguard mechanism via ROS-mediated inhibition of Tet3 catalytic activity. Neurite outgrowth assay showed that NACA treatment further

augmented the pro-growth phenotype of *Bmal1^cKO* (Fig. 5h). In aggregate, Bmal1 acts as an inhibitor of axon regeneration by limiting Tet3-5hmC gains; Bmal1 ablation mimics the conditioning lesion in lifting this epigenetic block partly in a Tet3-dependent manner, and the promoting effect can be potentiated by anti-ROS scavenger to unleash Tet3 catalytic activity (Fig. 5i).

## Tet3-5hmC regulon concerns axon growth, metabolism, and immune signaling

To identify the regenerative gene programs potentially impacted by Tet3-5hmC, we performed RNA-seq of naive DRG or conditioned DRG at 1 dpi (the same time point as in 5hmC mapping[17]), which identified 3,022 DEGs (cutoff: 0.25 log₂ fold change, *P* < 0.01) (Supplementary

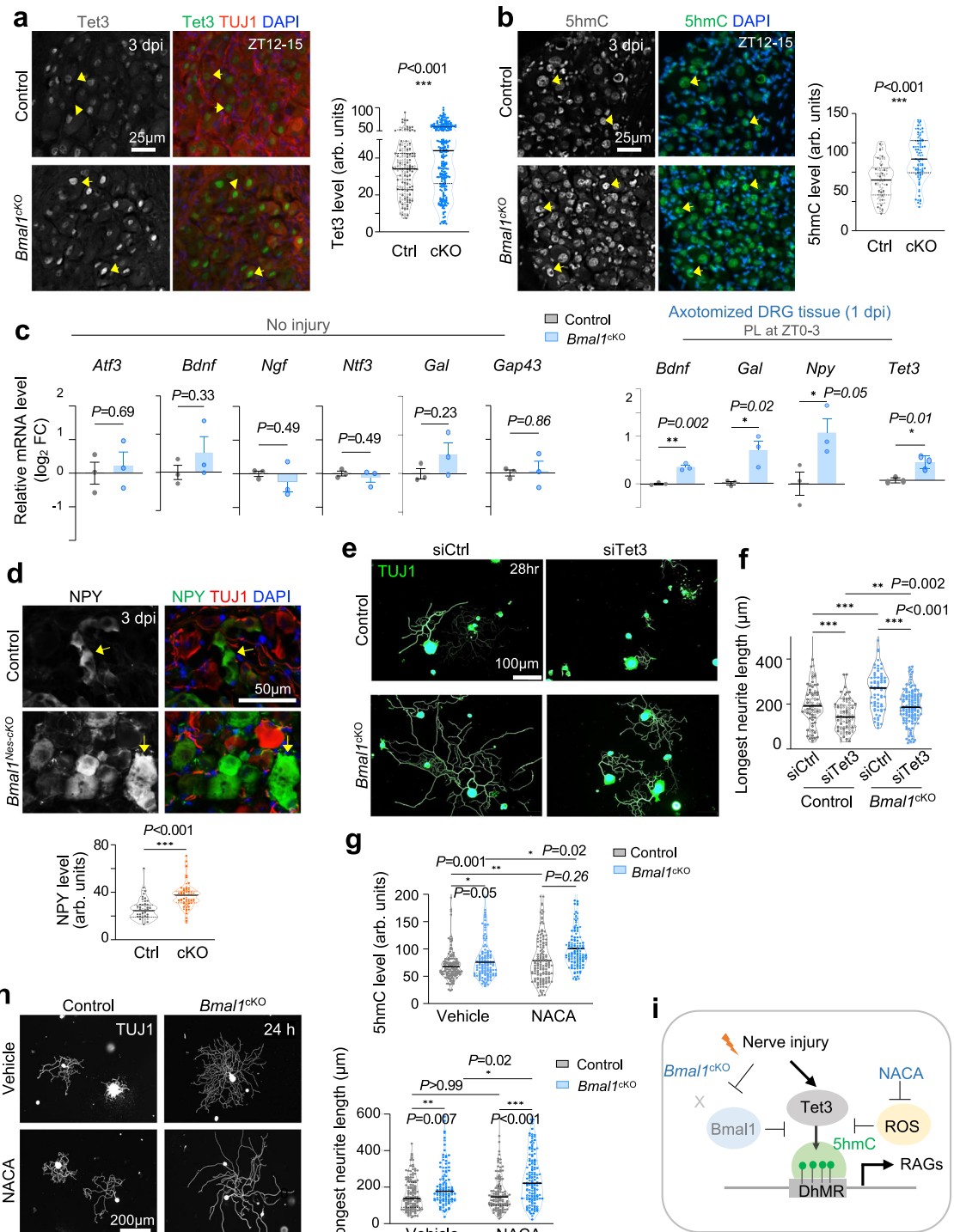

**Fig. 5 | Bmal1 affects Tet3-5hmC induction and RAGs expression after peripheral lesion of DRG.** Representative IF images of Tet3 (**a**) and 5hmC (**b**) expression in lumbar DRGs at 3 dpi after PL (ZT12–15). Violin plots of $n = 225$ control and 250 *Bmal1*cKO (Tet3) and $n = 90$ (5hmC) DRG neurons from $n = 9$ L4–6 DRGs of 3 mice per genotype. Median is shown by black line. Mann–Whitney test. **c** qRT-PCR of gene expression in lumbar DRGs with no injury (left) or PL at 1 dpi (right panel). *Bmal1* cKO augmented RAGs expression only after PL. $n = 3$ independent studies at ZT0–3, each with 6 L4–6 DRGs pooled from 2 mice. Data represent mean ± SEM. Unpaired two-tailed Student's *t*-test. **d** IF images of NPY expression in L4-L6 DRGs at 3 dpi after PL. $n = $ -70 neurons from $n = 9$ L4–6 DRGs of 3 mice per genotype. Median is shown by black line. Mann–Whitney test. **e**, **f** IF images of primary DRG neurons at 28 h after plating show that siTet3 reversed the growth advantage of *Bmal1*

deletion. $n = 70$–124 DRG neurons pooled from 2 mice per condition. Median is shown by black line. Two-way ANOVA followed by Bonferroni's multiple comparison test. **g** Quantification of 5hmC immunointensity in individual DRG neurons. Violin plots of $n = 150$–170 DRG neurons (L4–L6) from 2 mice for each condition. Black lines represent median. Two-way ANOVA followed by Bonferroni's multiple comparison test. **h** IF images of neurite outgrowth and quantification. Violin plots of $n = 115$–145 DRG neurons (L4–6) from 2 mice for each condition. Median is shown by black line. Two-way ANOVA followed by Bonferroni's multiple comparison test. **i** Model of Bmal1 and DNA hydroxymethylation in conditioned DRG neurons. Bmal1 deletion augments Tet3 induction, but elevated ROS inhibits Tet3 activity. NACA alleviates ROS inhibition, leading to further 5hmC gains and axon regeneration. ROS Reactive oxygen species, NACA N-acetyl-cysteine amide.

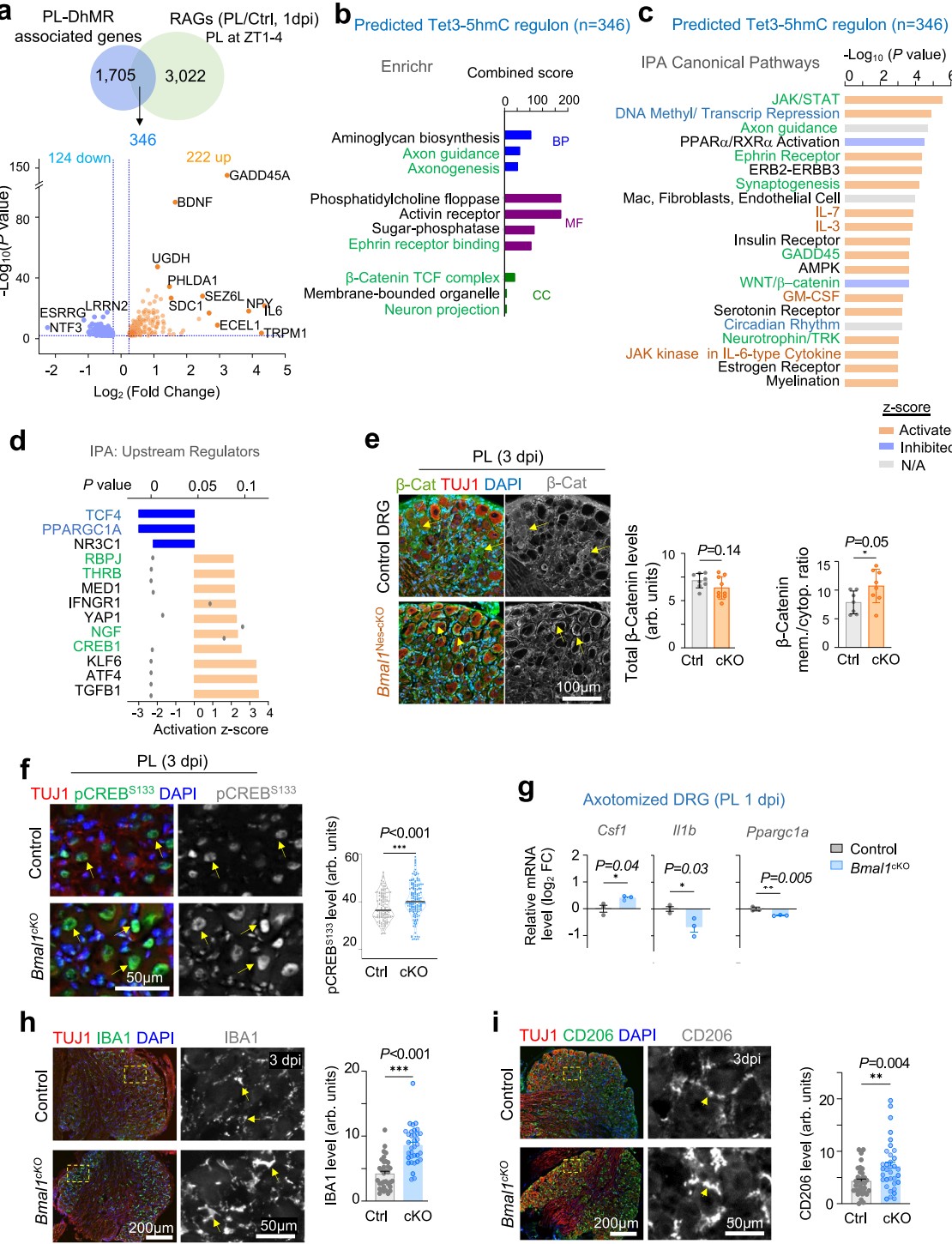

Data 4). Intersection with PL-DhMR associated genes revealed 346 overlapping genes, with 222 upregulated after PL, including well-established RAGs such as *Gadd45a, Bdnf, Npy*, and 124 downregulated (Fig. 6a; Supplementary Data 5). Enrichr analysis showed the predicted Tet3-5hmC regulon featured axon growth (e.g., Axon guidance, Axonogenesis, Neuron projection), metabolism (Aminoglycan Biosynthesis), and Wnt signaling (β-catenin TCF complex) (Fig. 6b). Ingenuity Pathway Analysis (IPA) further revealed that enriched pathways included Circadian Rhythm (verifying Bmal1 involvement), DNA Methylation/Transcriptional Repression (consistent with Tet3-5hmC epigenetics), GADD45 (linked to DNA demethylation and repair), Neurotrophin/TRK signaling, and notably immune signaling (IL-7, IL-3, JAK family kinase in IL-6-type Cytokine), all with a positive z-score,

while those with negative z-scores featured WNT/β-catenin and lipid metabolism (PPARα /RXRα activation) (Fig. 6c).

The upstream regulators of the Tet3-5hmC regulon as predicted by IPA with positive activation z-scores included CREB1, NGF, as well as RBPJ and THRB (the two TFs with binding motifs enriched in PL-DhMRs, see Fig. 1c), while those with negative activation z-scores included TCF4 (Wnt pathway), PPARGC1A (an integrator of energy metabolism and circadian clock[66]), and NR3C1 (a novel circadian regulator[67]) (Fig. 6d). For validation, we examined the expression of β-catenin, a transcription factor in the Wnt pathway and TCF4 interacting protein[68]. We found that at 3 dpi, while the overall expression levels appeared comparable, the membrane localization of β-catenin appeared higher in *Bmal1^cKO* DRG neurons than in controls, signifying

**Fig. 6 | Tet3-5hmC regulon governs axon growth and immune response after peripheral lesion of DRG. a** Top, Venn diagram showing the overlap of genes associated with PL-DhMRs ($n$ = 1705) and differentially regulated genes (DEGs) at 1-day post sciatic nerve crush injury compared to no injury (RAGs), identified by RNA-seq. Bottom, volcano plot of the 346 overlapping PL RAGs, with labeled top DEGs. Welch's two-sample $t$-test. **b** Enrichr analysis showing top enriched ontologies of the 346 RAGs identified in (**a**). **c** IPA showing the top enriched canonical pathways of the 346 RAGs. Bar color indicates predicted activation (orange), and inhibition (blue) of pathways, and gray bars indicate insufficient evidence for prediction of activation state. Right-tailed Fisher's exact test. **d** IPA analysis for top upstream transcriptional regulators of the 346 RAGs. Bar color indicates predicted activation (orange) or inhibition (blue) of pathways. Right-tailed Fisher's exact test. **e** Representative IF images and quantifications show higher membrane association of β-catenin in axotomized lumbar Bmal1$^{cKO}$ DRG neurons than in controls (TUJ1⁺) at

3 dpi after PL, but overall levels were comparable. Quantification of images from $n$ = 2-3 DRGs from 3 mice for each genotype. Data represent mean ± SEM. Unpaired two-tailed Student's $t$-test. **f** Representative IF images and quantifications show higher levels of nuclear pCreb$^{S133}$ in axotomized Bmal1$^{cKO}$ DRG neurons than controls (TUJ1⁺) at 3 dpi after PL. Violin plots of $n$ = 150 control and 170 Bmal1$^{cKO}$ neurons from $n$ = 8 DRGs from $n$ = 3 mice for each genotype. Median shown by black line. Mann-Whitney test. **g** qRT-PCR validation of the effect of Bmal1 ablation on gene expression related to immune or metabolism in axotomized lumbar DRGs at 1 dpi after PL. $n$ = 3 independent samples of L4–L6 DRGs, each pooled from 2 mice for each genotype. Data represent mean ± SEM. Unpaired two-tailed Student's $t$-test. Representative IF images and quantifications show increased expression of immune cell markers IBA1 (**h**) and CD206 (**i**) in axotomized lumbar DRGs at 3 d after sciatic nerve injury in Bmal1 cKO than control mice. $n$ = 17 DRGs collected from 6 pairs of mice. Data represent mean ± SEM. Mann–Whitney test.

inactivation of Wnt pathway, in line with bioinformatic results (Fig. 6e) and reported inhibitory role of Wnt in axon regeneration[69]. IF also verified that at 3 dpi, Bmal1$^{cKO}$ DRG neurons contained higher nuclear levels of pCREB$^{S133}$, a converging effector downstream of cAMP, MAPK, PI3K/ATK, and mTOR signaling (Fig. 6f).

We next analyzed PL-DhMRs with or without Bmal1 binding motif separately (Figs. S9a, S10a; Supplementary Data 6–8). For both categories, close to half displayed 5hmC gain, 40% 5hmC loss, and 10% both. The two groups of PL-DhMRs both showed increased chromatin accessibility after PL, as did the associated genes near TSS (see Fig. 1h and Fig. S10b). Of the predicted Tet3-5hmC regulon containing Bmal1 binding motif ($n$ = 129 RAGs, possibly direct targets of Bmal1) or not ($n$ = 223 RAGs, likely indirect targets of Bmal1 via Tet3), both shared enriched pathways for axon growth, circadian rhythm, and DNA methylation/transcriptional repression[69] (Figs. S9b, c, S10c, d), further supporting a link of circadian clock and genome-wide reconfiguration of DNA methylome after axonal injury. Top enriched pathways for those containing Bmal1 binding motif included HIF1α, mTOR, and autophagy (Fig. S9c). We further conducted IPA graphical summary analyses, which applies machine learning to prioritize and connect biological pathways and infer novel relationships[70]. This highlighted that the Tet3-5hmC regulon with Bmal1 binding motif featured cytoskeletal organization, formation of cellular protrusion, KLF6 (a known axon regeneration promoting TF[71–74]), while those without Bmal1 binding motif featured CREB1, IL1B, and immune response of macrophages (Figs. S9d, S10e). Of note, Bmal1 itself was part of the downregulated Tet3-5hmC regulon with Bmal1 binding motif, while Tet3 was part the regulon without Bmal1 binding motif, suggesting self-regulatory loops in response to PL (Supplementary Data 6 and 7).

## Neuronal Bmal1 ablation affects immune response in axotomized DRG

Immune pathways enrichment in the predicted Tet3-5hmC regulon in conditioning lesion implies potential non-autonomous effects of neuroepigenetics. We thus first surveyed the expression of immune-associated genes in uninjured DRGs. Consistent with only a priming effect of Bmal1 deletion, qRT-PCR showed no significant change of Ccl2, Il1b, Fcgr3a (encoding CD16), Mrc1 (CD206), Mki67 (proliferation marker) or Fn1 (matrix protein fibronectin) transcription in uninjured DRGs between genotypes (Fig. S11a). One exception was Il6, which was upregulated in mutant DRG. Since IL6 is a target of REV-ERB repressive activity[75], its upregulation aligned with the observed downregulation of REV-ERBs in Bmal1-deficient DRGs (see Fig. S5d). By contrast, conditioned Bmal1$^{cKO}$ DRGs at 1 dpi showed higher expression of Csf1 (colony stimulating factor 1, a potent regulator of macrophage activation), but lower expression of Il1b, while metabolism related gene Ppargc1a was slightly downregulated (Fig. 6g).

We observed higher levels of IBA1 (a marker upregulated in activated macrophages) and CD206 (a marker of pro-repair/anti-inflammatory macrophages) in conditioned but not uninjured Bmal1$^{cKO}$ DRGs

than control counterparts, but phagocytosis marker CD68 showed no overt changes between genotypes (Figs. 6h, i, S11a–c). Taken together, Bmal1 cKO appeared to affect neuroimmune interactions after axotomy.

## Bmal1 regulon concerns xenobiotic, lipid, and energy metabolism
To gain a more comprehensive view of Bmal1-regulated genes during axon regeneration, we performed additional RNA-seq on semi-purified primary DRG neurons (axotomized during dissociation) at 28 h after seeding. Principal component analysis (PCA) showed a clear segregation of three samples from Bmal1 cKO vs. control cohorts (both received tamoxifen) (Fig. 7a). We confirmed a significant down-regulation of Bmal1 in mutant samples (Fig. S12a). Hierarchical clustering revealed more upregulated than downregulated DEGs in Bmal1 cKO vs. control (Figs. 7b, S12b), in line with global Tet3-5hmC elevation and thereby transcriptional activation by Bmal1 ablation. The Bmal1 regulon in regenerating DRG neurons ($n$ = 625 DEGs, cutoff log$_2$FC of 0.25, $P$ < 0.01, Supplementary Data 9) included cytoplasmic proteins, membrane proteins, secreted factors, and nuclear factors (Fig. 7c), and they concerned not only axon growth (axon guidance, growth factor signaling), but also stress responses (e.g., Xenobiotic detoxification, Oxidative stress response, DNA damage repair, Autophagy, Unfolded protein response), lipid metabolism, and notably immune signaling (Fig. 7d).

To identify Bmal1 regulon potentially regulated via Tet3-5hmC epigenetics, we intersected the DEGs with PL-DhMR associated genes, yielding 85 overlapping genes (Fig. 7e; Supplementary Data 10). IPA of these genes showed positive activation z-scores for the canonical pathways of Phagosome formation, G-protein coupled receptor signaling, CREB signaling in neurons, HIF1α, and ID1 signaling, while Wnt/β-catenin showed a negative activation z-score (Fig. 7e, f). About 42% of them (36 genes) harbored Bmal1 binding motif (Supplementary Data 11), and they concerned axon growth (BDNF, Myc, RAC/PAK/p38), vascularization (HIF1α, VEGF, Angiopoietin receptor Tie2), as well as energy homeostasis and metabolism (Ghrelin regulation of energy homeostasis, Polyamine metabolism, Methionine De Novo and Salvage pathway, Vitamin D Receptor pathway) (Fig. 7g). Altogether, through Tet3-5hmC dependent and -independent mechanisms, Bmal1 regulates a large repertoire of genes related to axon guidance and growth, but also pathways not well studied in the context of conditioning lesion.

## Tet3 and 5hmC display diurnal rhythmicity in DRG neurons
Since Bmal1 is a core circadian regulator, we next asked whether expression of Bmal1 or Tet3-5hmC exhibited rhythmicity in DRG. We first compared two zeitgeber time (ZT) points: ZT0–3, corresponding to early resting phase for mice (ZT0: lights on), and ZT12–15, early active phase (Fig. 8a). qRT-PCR analysis revealed that Bmal1 transcription exhibited diurnal oscillation in DRG, lower at ZT12–15 than ZT0–3 (Fig. 8a). In contrast, Per1, Per2, Cry1, and Nr1d2, all negative

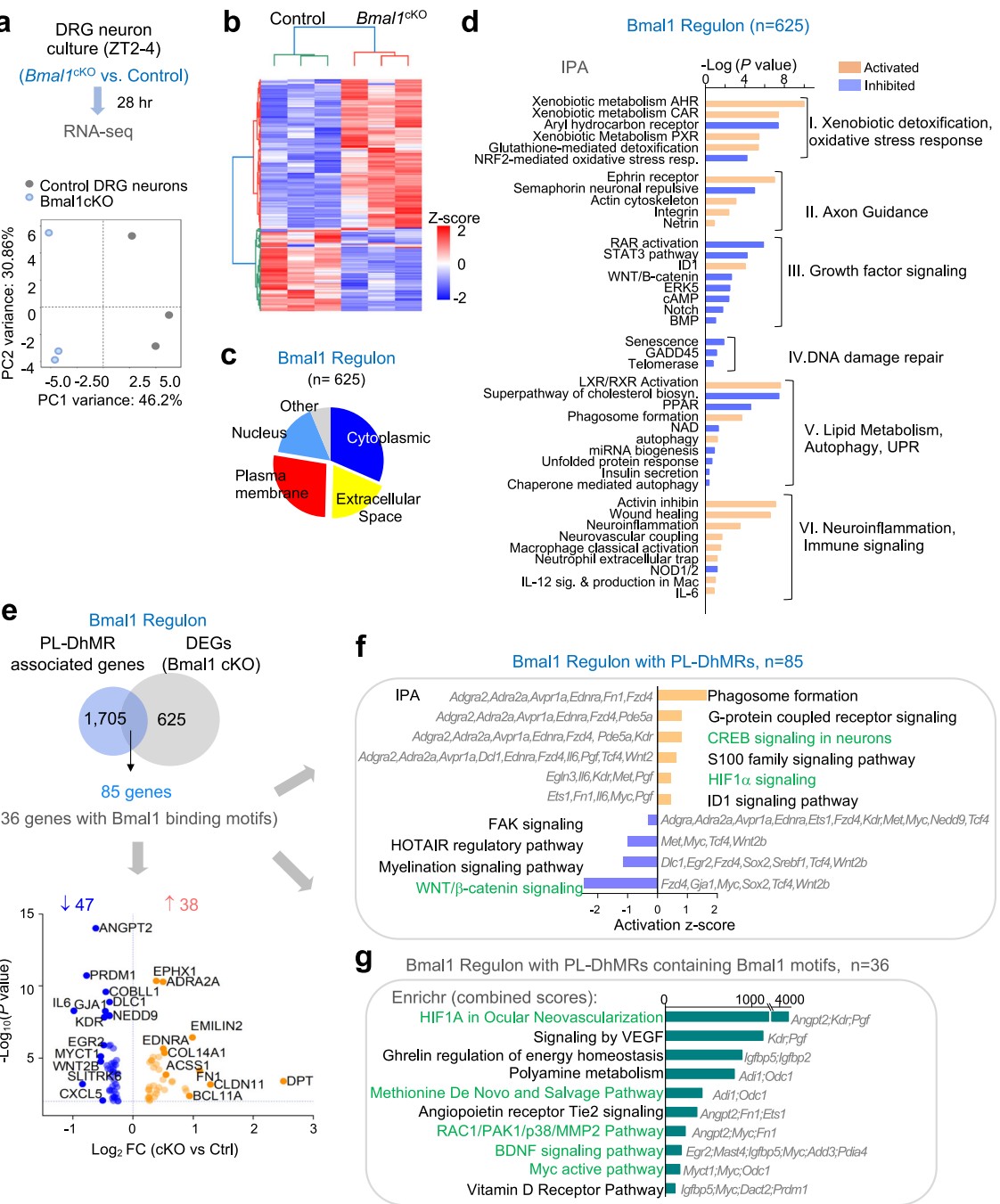

**Fig. 7 | Bmal1 regulon concerns stress responses and energy homeostasis during axon regeneration. a** Top, schematic of experimental paradigm for RNA-seq analysis on primary DRG neurons isolated from *Bmal1* cKO or control mice at ZT2–4 and cultured for 28 h. Bottom, principal component analysis of RNA-seq data shows segregation of samples from *Bmal1* cKO vs. control DRG neurons. **b** Heatmap of DEGs between *Bmal1* cKO and control neurons, cutoff: abs (log₂ FC) > 0.38 and *P* < 0.01. **c** Pie chart of subcellular localization of *n* = 625 Bmal1-regulated genes computed by IPA, cutoff: abs (log₂ FC) > 0.25 and *P* < 0.01. **d** IPA showing the top enriched canonical pathways of Bmal1 regulon, with color of bars indicating predicted activation (orange) or inhibition (blue) of pathways. Right-tailed Fisher's exact test. **e** Top, Venn diagram showing the overlap between genes associated with PL-DhMRs (*n* = 1705) and Bmal1 regulated genes (*n* = 625). Bottom, volcano plot of the 85 overlapping genes, with top DEGs labeled. **f** IPA highlights top canonical pathways of 85 Bmal1-regulated genes containing PL-DhMRs. Bar color indicates predicted activation (orange) or inhibition (blue). Pathway associated genes are labeled. Note, predicted activation of CREB and repression of Wnt/β-catenin signaling with *Bmal1* cKO. Right-tailed Fisher's exact test. **g**. Enrichr result of top pathways enriched in 36 Bmal1 regulated genes with Bmal1-motif in DhMRs.

feedback regulators of Bmal1, were transcribed at higher levels at ZT12–15 than ZT0–3, while *Clock* and *Nr1d1* did not show overt diurnal differences (Fig. 8a).

Resonating with a repressive role of Bmal1 on Tet3 expression, we observed diurnal changes of Tet3 and 5hmC in DRG neurons, higher at ZT12–15 than ZT0–3 (Fig. 8b–d). The time-of-day effect persisted for Tet3 expression in conditioned DRG neurons at 1 dpi, higher when

injury occurred at ZT12–15 (Fig. 8b, c). This also applied to 5hmC responsiveness to PL, with augmented 5hmC induction by PL at ZT12–15 than at ZT0–3 (Fig. 8d). We also examined transcriptional rhythmicity of the enzymes involved in DNA (de)methylation, and only *Tet3* and *Gadd45g* displayed diurnal changes (Fig. S13a, b), both containing promoter binding motifs for circadian clock TFs (Fig. S13c). Gadd45g is implicated in locus-specific DNA demethylation by serving

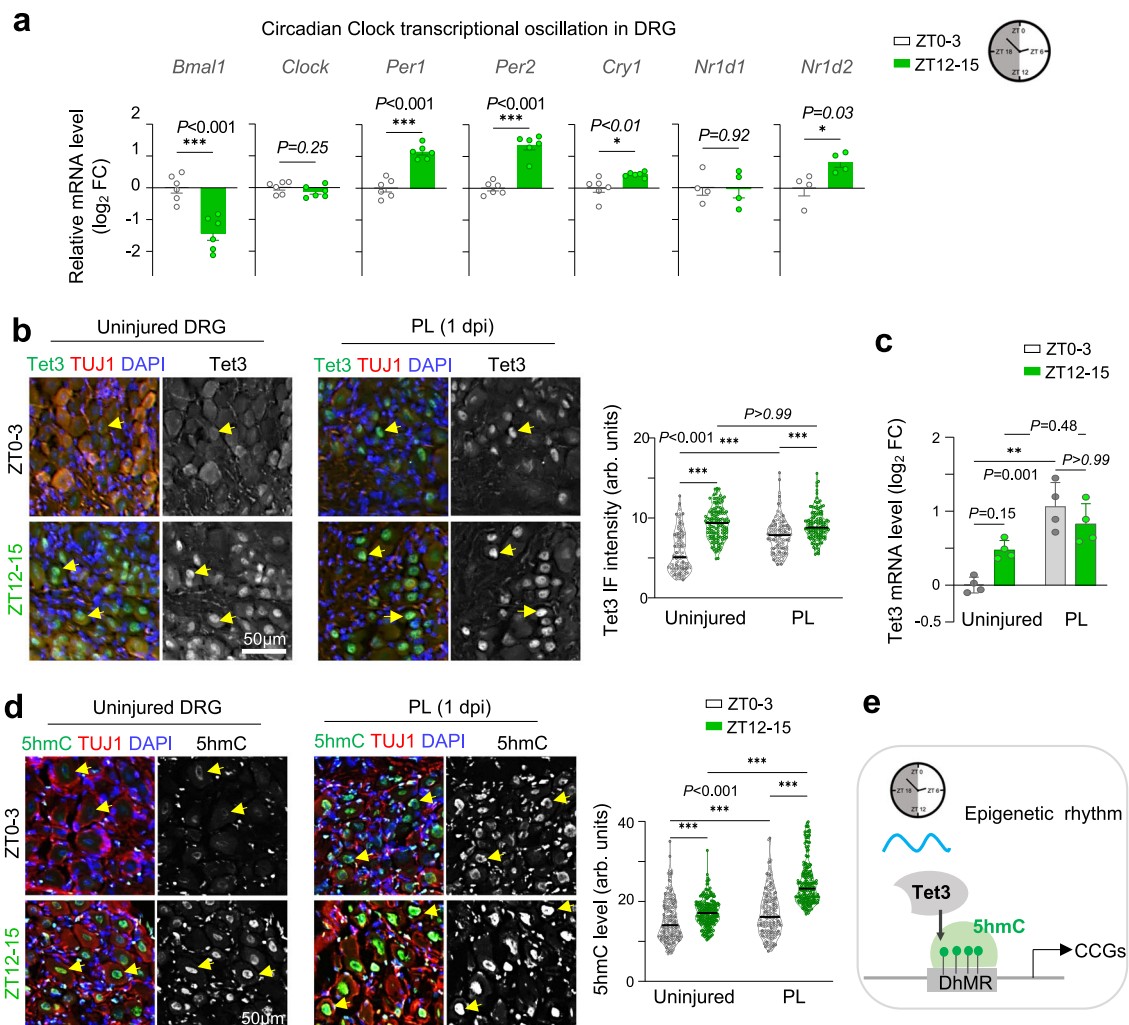

**Fig. 8 | Diurnal expression changes of Tet3 and 5hmC in DRG neurons. a** qRT-PCR reveals diurnal transcriptional changes of circadian clock genes in DRGs. Note the anti-phasic expression of *Bmal1* and circadian genes of the translation/transcription feedback loops that control Bmal1 rhythmicity. $n = 6$ independent samples for *Bmal1*, *Clock*, *Per1*, *Per2*, *Cry1* and $n = 4$ for *Nr1d1* and *Nr1d2*, each sample pooled from $n = 2$ mice. Samples were collected on consecutive days at the indicated ZT. Data represent mean ± SEM. Unpaired two-tailed Student's *t*-test. **b** Representative IF images and quantifications show higher nuclear levels of Tet3 in uninjured DRG neurons at ZT12−15 than ZT0−3. Note the time-of-day effect persisted on Tet3 upregulation after PL, with higher level at 1 dpi when injury occurred at ZT12−15. $n = 9$ DRGs isolated from 3 mice per condition. Data represent mean ± SEM. Two-way ANOVA followed by Bonferroni's multiple comparison test.

**c** qRT-PCR shows diurnal changes of *Tet3* transcription in uninjured DRGs, higher at ZT12−15 than ZT0−3. PL resulted in a robust *Tet3* induction for ZT0−3, and no further gain for ZT12−15 (which has a high baseline level of Tet3). $n = 4$ independent samples from 4 mice for each condition. Data represent mean ± SEM, Two-way ANOVA followed by Bonferroni's multiple comparison test. **d** Representative IF images and quantifications show diurnal changes of 5hmC in uninjured or axotomized DRG neurons (TUJ1⁺). Note more pronounced 5hmC elevation for PL at ZT12−15 than ZT0−3. $n = 9$ DRGs (L4−L6) isolated from 3 mice per condition. Median is shown by black line. Two-way ANOVA followed by Bonferroni's multiple comparison test. **e**. Schematic model of epigenetic rhythm of Tet3-5hmC in DRG neurons.

as an adapter for BER factors[76]; its higher expression level at ZT0−3 thus aligns with lower 5hmC accumulation at this ZT phase.

We further examined rhythmicity of Tet3-5hmC expression in DRG neurons by performing IF at 4-hour intervals along the ZT cycle, which revealed zenith of Tet3 levels at ZT20 and 5hmC at ZT12, while nadir of Tet3 expression at ZT0 and 5hmC at ZT4 and ZT20 (Fig. S14a). Interestingly, Bmal1 protein levels as assessed by IF appeared higher at ZT12 than ZT0, opposite that of *Bmal1* transcriptional oscillation, echoing reported lag time between Bmal1 mRNA and protein levels[77,78] (Fig. S14b). In this context, it is noteworthy that Bmal1 protein level does not equate to Bmal1 DNA binding or transcriptional activity as Bmal1 activity is additionally controlled by regulators of the feedback loops such as Per and Cry[47,77,79]. Indeed, highest Bmal1 activity has been reported when Bmal1 protein level is at its lowest[79]. Of note, in whole DRG lysates (from either male or female mice), WB did not reveal significant changes of Bmal1 protein levels between ZT0−3 and

ZT12−15, although we detected a slight shift to higher molecular weight at ZT12−15 (Fig. S14c), likely reflecting post-translational modifications[79,80]. We also performed WB on DRG tissue lysates collected at ZT0−3 from *Bmal1*$^{cKO}$ or control mice, which revealed only a slight but not statistically significant reduction of Bmal1 in mutant samples, likely due to neuronal-specific cKO that spared glial cells (Fig. S14d).

To understand whether the circadian rhythm in DRG may be connected to axon growth potential, we first leveraged a recent RNA-seq study on CCGs in DRG (Fig. S15a; Supplementary Data 12)[81]. We found that CCGs in DRG ($n = 832$) were enriched for WNT/β-catenin, Axon guidance, PTEN, CREB, Rac, Calcium signaling, as well as neuronal circuit homeostasis (Neuregulin, Synaptogenesis) and immune activity (IL-15 production, Phagosome formation, Neuroinflammation, Macropinocytosis) (Fig. S15b). We validated diurnal patterns of nuclear pCREB$^{S133}$ and membrane associated β-catenin in DRG neurons by IF,

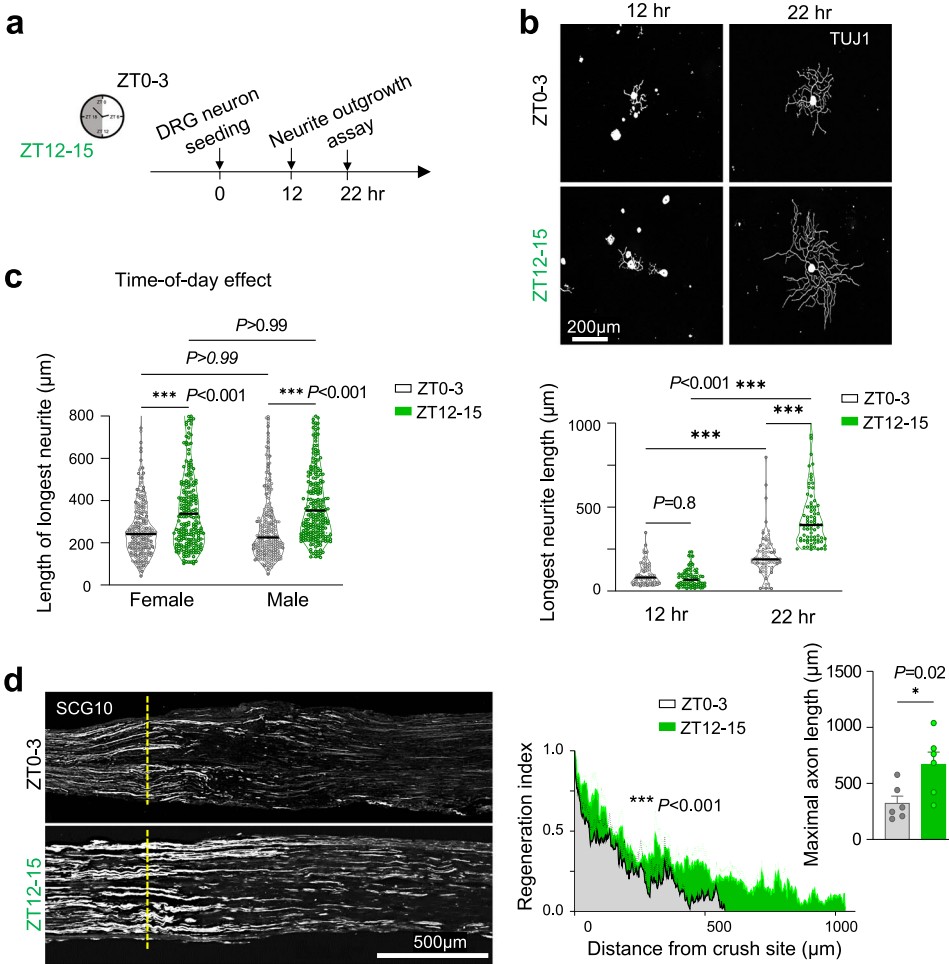

**Fig. 9 | Time-of-day effect on axon regeneration after conditioning lesion.**
**a** Experimental scheme of neurite outgrowth assay with primary DRG neurons isolated at ZT0–3 or ZT12–15. **b** Representative IF images and quantification show DRG neurons from ZT12–15 extended longer neurites than those from ZT0–3 at 22 h but not 12 h in vitro. Violin plots of $n = 77$ neurons, pooled from 3 independent cultures isolated from 3 mice per group. Median shown by black line. Kruskal–Wallis with Dunn's multiple comparison's test. **c** Quantifications show time-of-day effect on neurite outgrowth from DRG neurons isolated from both male and female mice. $n = 250$ neurons from 3 mice per group. Median shown by black line. Two-way ANOVA followed by Bonferroni's multiple comparison test. **d** Representative IF images and quantifications show faster axonal regeneration (SCG10⁺) at 1 d after sciatic nerve crush injury at ZT12–15 compared to ZT0–3. For regeneration index, Two-way ANOVA. $n = 6$ mice per group. For maximal axon length, data represent mean ± SEM. Unpaired two-tailed Student's $t$-test.

both appeared higher at ZT12–15 than ZT0–3 (Fig. S15c, d). Moreover, we confirmed diurnal transcriptional rhythm of axon guidance molecules and repair-related genes (*Bmp2, Smad4, Sema5a, Fn1, Mki67,* and *Col4a1)* in uninjured DRGs (Fig. S15e).

### Time-of-day effect on axon regrowth potential in vitro and in vivo

Having revealed diurnal oscillation of Bmal1 and Tet3·5hmC in DRG neurons (Fig. 8e), we next examined a potential time-of-day effect on axon growth potential by comparing primary DRG neurons collected at ZT0–3 and ZT12–15 (Fig. 9a). While the initial phase of axon elongation at 12 h after plating did not show differences, by 22 h, DRG neurons collected at ZT12–15 extended longer neurites than those from ZT0–3 (Fig. 9b). This applied for DRG neurons from both female and male mice, indicating a sex-independent diurnal effect on axon growth (Fig. 9c). We further conducted neurite outgrowth assay on primary DRG neurons isolated every 6 h, wherein the ones isolated at ZT12 extended the longest neurites while those isolated at ZT0 extended the shortest neurites (Fig. S16a).

We next conducted in vivo nerve regeneration studies. In congruence with the neurite outgrowth results, axonal regrowth appeared more robust for PL occurring at ZT12–15 than at ZT0–3, with SCG10⁺

axon length reaching twice as long at 1 dpi (Fig. 9d). Altogether, axonal injury at the onset of the active phase of animals appears to be associated with more robust axon regrowth.

### *Bmal1* cKO mimics the conditioning lesion in augmenting axon growth potential at different ZT

Our above studies showing that PL triggers *Bmal1* downregulation and that *Bmal1* deletion leads to enhanced axon regeneration were conducted mostly at ZT0–3. This prompted us to ask if these results would be affected by different ZTs (Fig. 10a). We found that *Bmal1* downregulation at 1 dpi was more pronounced for PL at ZT12–15 than ZT0–3 (Fig. 10a, b). Consistently, conditioned DRGs with a prior PL at ZT12–15 extended longer axons than the ZT0–3 group (Fig. 10c), indicating a time-of-day influence on the conditioning lesion effect, more pronounced at the onset of active phase of animals.

Perhaps reflecting light entrainment or "masking", a response to light overriding the absence of circadian clock[34,52,82], DRG neurons from *Bmal1^cKO* mice (maintained in regular light-dark (LD) cycle) continued to show time-of-day effect on neurite outgrowth, with longer neurites at ZT12 than ZT0 (Fig. S16b). Likewise, neurite outgrowth advantage of *Bmal1^cKO* also appeared more pronounced for the ZT12–15 group than ZT0–3 (Fig. 10d).

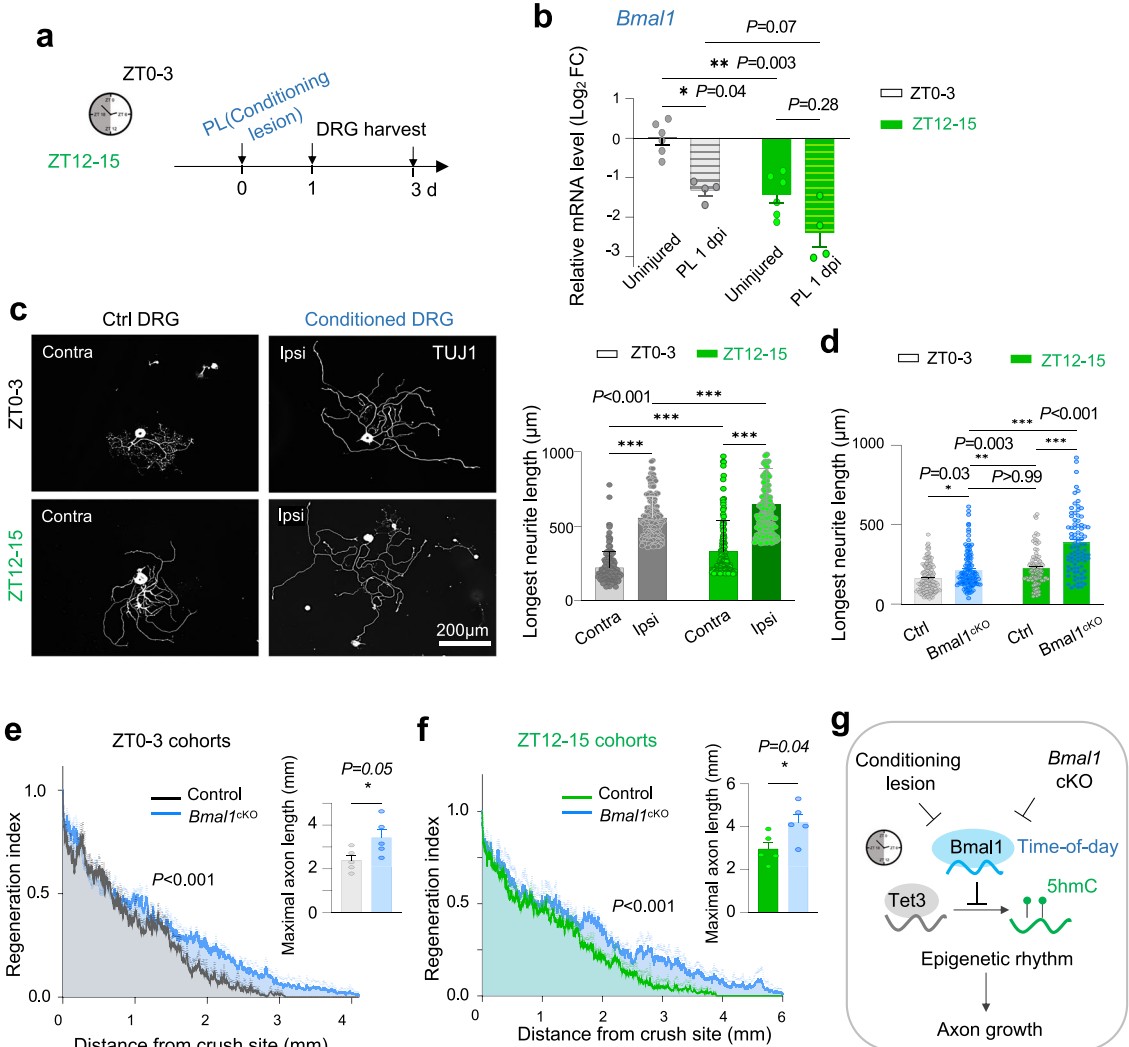

**Fig. 10 | Time-of-day impact on conditioning lesion effect for axonal regrowth.**
**a** Experimental scheme to evaluate time-of-day effect on the conditioning effect
from PL. Mice were subjected to sciatic nerve lesion (PL) at ZT12–15 or ZT0–3, and
DRGs were dissociated at 1 dpi for neurite outgrowth assay. **b** qRT-PCR shows that
*Bmal1* downregulation at 1 dpi was more pronounced when the sciatic nerve injury
occurred at ZT12–15 than ZT0–3 (uninjured DRG data were also shown in Fig. 8a).
Each sample pooled from 3 ipsilateral lumbar DRGs from $n = 4$ mice per group. Data
represent mean ± SEM. Two-way ANOVA followed by Bonferroni's multiple com-
parison test. **c** Representative IF images and quantification of neurite outgrowth of
DRG neurons 24 h after plating. A conditioning lesion was performed 1 day prior at
either ZT12–15 or ZT0–3. Quantification of 149 neurons collected from $n = 4$ mice
per group. Data represent mean ± SEM. Two-way ANOVA followed by Bonferroni's

multiple comparison test. **d** Quantification of time-of day effect on neurite out-
growth from *Bmal1cKO* or control DRG neurons. $n = 115$–135 DRG neurons for ZT0–3
and $n = $ -90 for ZT12–15 per group (ZT0–3 for control mice were shown in Fig. 5h).
Data represent mean ± SEM. Two-way ANOVA followed by Bonferroni's multiple
comparison test. Quantifications of in vivo axon regrowth show that *Bmal1* cKO
resulted in enhanced axon regeneration (SCG10+) at 3 dpi for ZT0–3 (**e**) and
ZT12–15 (**f**) injury groups. Two-way ANOVA. Bar graphs of maximal axon length are
shown on right. Data represent mean ± SEM. Unpaired two-tailed Student's *t* test.
$n = 5$ mice per genotype. **g** Working model of how *Bmal1* ablation mimics the
conditioning lesion in lifting the circadian block, leading to enhanced axon growth
capacity, in part through augmented Tet3-5hmC reconfiguration.

Resonating with the above results, when we examined in vivo
axon regrowth after sciatic nerve crush injury at different ZTs, the axon
promoting effect from *Bmal1* ablation was evident for both ZT cohorts,
but more pronounced for the ZT12–15 cohort, with maximal axon
length reaching 4.2 ± 0.39 mm at 3 dpi for ZT12–15 vs. 3.4 ± 0.84 mm
for ZT0–3 (Fig. 10e, f).

Since the conditioning lesion operates partly through CREB
activation[83,84] and p-CREB[S133] expression displays diurnal changes in
uninjured DRG neurons (see Fig. S15c), we next examined p-CREB[S133]
levels in axotomized DRG neurons, which showed higher levels at
ZT12–15 than ZT0–3 for both 1 dpi and 3 dpi (Fig. S16c, d). Aligned
with highly elevated p-CREB[S133] in *Bmal1cKO* DRGs (see Fig. 6f),
treatment with db-cAMP, a cell permeable synthetic analog of cAMP
which activates CREB, did not further enhance neurite outgrowth

(Fig. S16e). Hence, while DRG neurons display diurnal changes of
axon growth capacity, attenuating Bmal1 by either conditioning
lesion or genetic ablation can further augment axon regeneration
even for ZT12–15 (Figs. 10g, S16f).

Lastly, as another indicator of the time-of-day effect, we com-
pared the DEGs profiled in conditioned DRG with PL performed at
ZT0–3 or at undefined ZT time[14,85] (Fig. S17a). This revealed
815 shared genes with similar directionality of expression (Pearson
correlation coefficient of 0.89) and featuring Circadian Rhythm,
Axonal guidance, cAMP-mediated pathway, CREB, HIF, and Xeno-
biotic metabolism, as well as immune signaling (Fig. S17b, c; Sup-
plementary Data 13). But notably, there was a large fraction of non-
overlapping DEGs: 73% of the DEGs from this study and 58% of the
published study[14,85].

## Discussion

Axon regeneration requires epigenetic reconfigurations to rewire the gene regulatory circuits. Here, we identified Bmal1 as a regulator of axon regeneration: *Bmal1* deletion led to enhanced axon regeneration, in part through augmenting Tet3-5hmC induction in response to axotomy. We also revealed diurnal oscillation of Tet3-5hmC in DRG and a time-of-day effect on axon growth potential.

Peripheral axotomy of DRG induces Tet3 expression to adjust epigenetic states in favor of axon regeneration[17]. Our unbiased survey of the PL-specific DhMRs revealed an enrichment of Bmal1 binding motif, thus implicating a link of Bmal1 with epigenetic regulation of axon regeneration. Interestingly, our data supported an antagonistic relationship between Bmal1 and Tet3 in gating axon regenerative responses: (i) rapid downregulation of Bmal1 after conditioning lesion of DRG, coinciding with Tet3-5hmC upregulation; (ii) augmented Tet3 upregulation and 5hmC gains with *Bmal1* ablation. Tet3 may be under the direct control of Bmal1-Clock or other molecular oscillators (*Tet3* promoter contains binding motifs for Clock and Bhlhe41). Tet3 recruitment to PL-DhMRs may also involve Bmal1 or other bHLH-PAS TFs. Alternatively, a recent study revealed that the Tet3 catalytic domain displays an intrinsic sequence preference for E-box motifs to establish DNA (de)methylation patterns;[86] thus, by occupying the E-box motifs, Bmal1 may sterically restrict access of Tet3 to these sites. The potential dual mechanism of limiting Tet3 expression and controlling Tet3 access to E-box motifs may ensure swift and robust gene regulation by the Bmal1 circadian machinery in response to axotomy. While our data demonstrated a transcriptional as well as a functional link of Bmal1 and Tet3 (as the growth advantage of Bmal1-deficient DRG neurons requires Tet3), future analysis of a possible protein-protein interaction is needed.

By enhancing Tet3 expression and global 5hmC gains, *Bmal1* deletion likely exerts a wide-spread influence on the DNA methylome and thereby transcriptional output. This may also underlie how Bmal1 operates as a pioneer-like TF[38,39] in the setting of axon regeneration. A pioneer-like function of Bmal1 for Tet3-5hmC epigenetic reconfigurations also fits with our findings that *Bmal1* deletion alone seems insufficient to trigger large-scale pro-growth gene programs, but it primes DRGs for regenerative responses. The signal that triggers RAGs induction upon axotomy awaits future study, but may involve axonal retrograde transport of injury signals back to soma[87].

While global 5hmC gains were observed with *Bmal1* deletion, on the level of individual DhMRs and associated genes, either 5hmC gain or loss may occur, depending on the step along the DNA demethylation cascade[88,89]. Of note, although DNA methylation primarily occurs near TSS and generally leads to gene silencing[90], 5hmC modifications typically occur at introns and intergenic regions rather than at promoters and CpG islands[17], and may be associated with neuronal enhancer regions[91,92]. In congruence, CpG island DNA methylation analyses revealed only minor changes after PL vs. CL of DRG[15].

How *Bmal1* is downregulated after PL requires further study. In this regard, it is notable that *Per1* is upregulated after PL; since *Per1* is an immediate early gene[93], it may participate in further suppressing Bmal1 transcriptional activity after PL. The numerous feedback loops of the circadian clock preclude a clear demarcation of the hierarchy of Bmal1 and Tet3 regulation at this point. Indeed, both *Tet3* and *Bmal1* genes contain PL-DhMRs, suggesting auto-regulatory loops and sensitivity to DNA methylation changes.

Our transcriptional profiling revealed a large repertoire of Bmal1-regulated genes in conditioned DRG, with partial overlap with PL-DhMR associated genes. Hence, Bmal1 operates through Tet3-dependent and -independent mechanisms in response to axotomy. It is noteworthy that the Bmal1 regulon not only concerns axon growth/axon guidance, but also immune signaling, stress responses, lipid metabolism, and energy homeostasis, thereby pointing to additional pathways with potential impact on axon regeneration. Intriguingly, 46% of the Bmal1 regulon in injured DRG neurons encoded proteins located at plasma membrane or extracellular space, supporting non-autonomous interactions. Also of interest is whether Bmal1 activity in glial cells influences regeneration; in non-CNS, the beneficial effects of Bmal1 deletion for tissue repair have been attributed to immune or vascular changes[94–97].

Circadian rhythm is governed not only by the central clock located in the suprachiasmatic nucleus, but also by interactions with peripheral clocks, which exist in every organ system including DRGs[81,98]. In our studies, we utilized the SLICK-H Thy1-CreER[T2] line for *Bmal1* cKO, which may potentially ablate Bmal1 in the suprachiasmatic nucleus, affecting the central clock. However, earlier studies using constitutive deletion of *Bmal1* showed that behavioral rhythms and activity are maintained in knockout mice entrained by light:dark cycle[34,52,82]. As our mouse cohorts were kept in regular light:dark cycle, we expect that perturbation of the central clock is not a major factor for the observed phenotypes of axon regeneration in *Bmal1*^cKO animals. Bmal1 notably also has circadian-independent functions, e.g., in aging and tissue homeostasis[99].

The circadian clock affects a wide range of systems, including metabolism, inflammation, and stress responses[100]. Our study uncovered Tet3-5hmC rhythmicity in connection with a time-of-day effect on axon regrowth potential. In echo, recent studies revealed changes of cytosine modifications in a circadian manner in mouse liver and lung, particularly at E-box elements of circadian transcripts in gene body region[101,102]. Tet3-5hmC diurnal changes in DRG neurons may be controlled not just by peripheral circadian clock, but also reflect light entrainment, sleep cycle, or neuronal activity. In the current study, the *Bma1*^cKO mice were maintained in regular light:dark cycle, thus the continued time-of-day differences of axon growth potential of DRG neurons from *Bmal1*^cKO mice might reflect "masking" or that the glial circadian clock remained intact in the cKO mice. To tease out whether the Tet3-5hmC epigenetic oscillation or axon growth potential is truly circadian or a result of light entrainment or neuronal activity requires further studies in animals kept in constant darkness. In addition, increasing neuronal activity by electrical stimulation or by exercise can improve regeneration[103]; since exercise affects circadian rhythms and alters the amplitude of the clock[104], its regenerative effect may partly operate through circadian modulation. It is also worth future exploration if other epigenetic mechanisms such as histone modifications display diurnal rhythmicity. Since conditioned DRG neurons extended longer axons than *Bmal1*^cKO neurons, Bmal1 circadian rhythm is only one factor at play for axon regeneration.

From an evolutionary point-of-view, faster initiation of the regenerative response in active phase may reflect an anticipation by neurons for heightened regenerative need during the active phase. Alternatively, gene expression program at the active phase may simply coincide with a large set of genes controlling axon regeneration. Since most rodent experiments are conducted during daytime with lights on (resting phase of rodents), our study underscores the importance of experimental setups in a circadian-informed manner[105,106].

In sum, our study identifies a role of the circadian oscillator Bmal1 in the regulation of neural repair and provides a proof of principle that suppression of circadian TF such as Bmal1 can foster a sustained regenerative injury response. As a growing arsenal of pharmacological agents targeting circadian pathways are available[107], the understanding of the connection between Bmal1 and Tet3/5hmC epigenetics, as well as regenerative and stress responses, provides a starting point to explore strategies to enhance axon regeneration.

## Methods

### Mouse strains

Animal procedures were conducted according to the protocol (IPROTO202200000184) approved by the Institutional Animal Care and Use Committee (IACUC) at Mount Sinai. C57BL/6J (JAX stock

#000664), Tg(Thy1-cre/ERT2,-EYFP)HGfng (also known as SLICK-H; stock #012708)[54], B6.Cg-Tg(Nes-cre)1Kln/J (stock #003771)[108], and *Bmal1*[fl/fl] (stock #007668)[109] mice were obtained from The Jackson Laboratory. All mice were bred onto C57BL/6J genetic background for at least three generations. Animals were housed under 12:12 h light:-dark cycle with ad libitum access to food and water and were habituated to the facility for at least 2 weeks prior to conducting experiments. Tamoxifen (Sigma) dissolved in corn oil (Sigma) was administered at a dose of 100 mg/kg by intraperitoneal injection once a day for 5 consecutive days. Experiments were conducted 2 weeks after tamoxifen injections and within the first 3 h of light-on in animal facility (ZT0–3) unless otherwise specified.

Genotyping was performed by PCR using DNA prepared from tail tips or DRGs with the following primers:

For *Bmal1* floxed allele: L1: ACTGGAAGTAACTTTATCAAACTG, L2: CTGACCAACTTGCTAACAATTA, R4: CTCCTAACTTGGTTTTTGTCTGT; floxed allele band size is 431 bp, excised allele 570 bp, and wild-type allele 327 bp.

For Thy1-Cre[ERT2/EYFP] (SLICK-H): Thy1F1: TCTGAGTGGCAAAG-GACCTTAGG, EYFPR1: CGCTGAACTTGTGGCCGTTTACG, Int-F: CAAA TGTTGCTTGTCTGGTG, Int-R: GTCAGTCGAGTGCACAGTTT; Transgene band is ~200 bp and internal positive control 324 bp.

For Nestin-Cre: Nestin-F: CCGCTTCCGCTGGGTCACTGT, Nestin-R: TGAGCAGCTGGTTCTGCTCCT, Cre-R: ACCGGCAAACGGACAGAAG CA; transgene band is 229 bp and wild-type band 379 bp.

## Sciatic nerve lesion injury

Sciatic nerve crush injury was performed on anesthetized mice with isoflurane under aseptic conditions. After shaving the leg fur and sterile prepping, a skin incision was made on right mid-thigh region, and the fascia between the gluteus superficialis and biceps femoris muscles was gently opened to expose the sciatic nerve, which was then carefully freed from surrounding connective tissue. Crush injury was performed approximately 5 mm distal to the bifurcation point of the sural nerve, applying pressure with ultra-fine hemostats (FST, #13020-12) for 15 seconds. Wound area was closed with two reflex 9 mm clips (Braintree Scientific, EZC-2PK). Sciatic nerve transection injury was performed with 3 mm Vannas spring scissors (FST, #15000-00).

## DRG isolation procedure

Dissections were performed under a Nikon SMZ645 stereo microscope. Mice were euthanized by carbon dioxide inhalation followed by cervical dislocation. Mice were laid on the dorsal side and immobilized on a dissection pad. Skin covering the ventral thorax and abdomen was removed together with internal organs to fully expose the ventral spinal column using surgical scissors (Fine Science Tools, #14054-13), supported by tissue forceps (Fine Science Tools, #11021-12). Muscles covering ventral side of the spinal column were removed using Spring Scissors (Fine Science Tools, #15751-11) to expose the lumbosacral peripheral nerves. To expose the lumbosacral DRGs, the ventral vertebrae were removed using the aforementioned Spring scissors, aided by Octagon forceps (Fine Science Tools, #11042-08) with care taken not to sever nerves. DRGs were gently picked starting with the lower lumbar level (L6) using a Dumont #3 forceps (World Precision Instruments, #50037) and isolated by cutting connecting nerves with Vannas Spring scissors (Fine Science Tools, #15000-00).

## Lentiviral production and DRG transduction experiments

HEK293T producer cells were subjected to multiple passaging steps and then seeded at density of $1 \times 10^6$ in a T75 flask containing 15 ml of complete media, DMEM (Invitrogen, #10569044), 1× NEAA (Invitrogen, #11140-050), and 10% FBS. Two days later, $3.8 \times 10^6$ cells were seeded in PLO coated 10 cm plate. Next day, cells were washed with DPBS and incubated with 10 ml OPTI-MEM (Invitrogen, #31985070). Transfection mix containing 1.3 pmol psPAX2 (Addgene #12260), 0.72 pmol

pMD.2G (Addgene #12259), and 1.64 pmol pEGIP or Bmal1-OE plasmids and PEI (1:3 µg DNA to PEI ratio) was added to cells for 6 h. Cells were washed twice with DPBS and incubated in complete media. Media containing the virus was collected at 48 h, 72 h, and 96 h post-transfection. To concentrate virus, media was spun at 500 g for 5 min, filtered through 0.45 micron PES filter (Thermofisher, #725-2545), and spun at 24,000 rpm for 2 h using SW41 Beckman rotor. Media was aspirated carefully and pellet containing virus was suspended in 100 µl serum free DMEM per 10 ml initial volume. Virus was stored at 4 °C and used within 1 week of preparation.

For viral transduction of adult DRG neurons, neurons freshly isolated from thoracic and lumbar DRGs of one mouse were suspended in 1 ml of 2x complete DRG media without antibiotic/antimycotic. 30 µl virus was mixed with 970 µl Neurobasal-A media and added to the 1 ml neuronal suspension. The resulting 2 ml neuron-virus mix was added to a 6-well plate precoated with PLO/laminin, spun down at 900 rpm for 90 min at room-temperature, and incubated at 37 °C overnight. Next day, media was changed to 1x complete DRG media with antibiotic and antimycotic. For neurite outgrowth analysis, transduced DRG neurons were washed once with DPBS, trypsinized for 3 min at 37 °C, and resuspended in DRG media by gentle pipetting. Neurons were split from one 6-well plate to six wells of a 24-well plate with PLO/laminin precoated coverslips for analysis.

## DRG neuron culture

For culture of primary DRG neurons, isolated DRGs were placed in DMEM/F12 solution on ice, washed 3x with ice-cold HBSS-HEPES buffer, and incubated in 3% collagenase (Worthington, #LS004196) for 90 min at 37 °C. Subsequently, collagenase was removed and DRGs were washed three times with HBSS-HEPES at room-temperature and incubated with 0.25% trypsin-EDTA containing 50 mg/ml DNase I (Worthington, #LS002138) for 30 min at 37 °C. Trypsin was removed and DRGs were triturated in 0.5 ml medium containing DMEM high-glucose Glutamax (Gibco, #10569-044), 10% fetal bovine serum, 1× nonessential amino acids, and 50 mg/ml DNase I by pipetting fifteen times through an unpolished and fifteen times through a fire-polished pasture pipette. Triturated DRGs were suspended in 8 ml of Neurobasal-A media and cleared from myelin debris using 2 ml of 5% BSA in HBSS-HEPES as cushion. Finally, neuronal pellet was resuspended in DRG medium containing 2% B27 (Gibco, #12587-010), 0.5 mg/ml glucose, 0.5 mM L-glutamine, 0.4% antibiotic-antimycotic agent (Gibco, #15240062) in Neurocult NB-A media (Stemcell technologies, #5750). For seeding of DRG neurons, glass coverslips (Fisher Scientific, #50-143-822) were preprepared by inserting coverslips in a 24-well plate and coating them with 30% poly-L-ornithine (Sigma, P4957) at 37 °C for 1 h. Coverslips were washed two times with PBS (with calcium and magnesium), and coated with 10 mg/ml laminin (Gibco, #23017-015) in DMEM/F12 (Gibco, #11330-057) for 1 hr at 37 °C. Before seeding of neurons, slides were washed once with Neurocult NB-A media. For neurite outgrowth assay, ~1200 DRG neurons were seeded per cm² and analyzed at the indicated times. Pan-Tet inhibitor C35 (Aobious, AOB11121) was used at a concentration of 8.3 µM in DMSO. N-acetylcysteine amide (NACA, Selleckchem S5804) and db-cAMP were freshly prepared and used at 400 µM and 1 mM, respectively. Circadian drugs, KL001 (Cayman, #13878) or PF-670462 (Cayman, #14588) were used at 5 and 1 µM, respectively. Both drugs were prepared in DMSO and stored at −80 °C at stock concentrations of at least 50 mM. siRNA-mediated knockdown studies were performed as previously described with slight modification[110]. Briefly, for each condition, ~4000 DRG neurons were resuspended in 1.5 ml of titration media (without DNase I) and gently mixed with 0.5 mL of transfection complex containing 2 µl of DharmaFECT 2.0 (Dharmacon, #T-2002-02) and 2 µl of siRNA at 20 µM stock concentration in Neurocult NB-A media. Cells were seeded on PLO/laminin coated plates and analyzed as indicated. ON-TARGETplus SMART pool siRNA oligos were ordered

from Dharmacon (siTet3, L-054156-0005 and non-targeting pool, D-001810-10-05).

## DRG explant culture

DRGs were isolated from adult mice and placed individually in DMEM/F12 solution on ice, washed twice with ice-cold HBSS-HEPES buffer, and incubated in 9% collagenase (Worthington, #LS004196) for 30 min at 37 °C. Subsequently, collagenase was removed and DRGs were washed twice with HBSS-HEPES. DRGs were picked and placed individually on laminin-coated coverslips in 0.5 ml medium containing DMEM high-glucose with Glutamax (Gibco, #10569-044), 10% fetal bovine serum, and 1× non-essential amino acids. A centrifugation step at 500 × $g$ for 5 min was used to promote adherence. Explants were cultured at 37 °C for 4 days followed by fixation with 4% paraformaldehyde containing 4% sucrose for 20 min for immunofluorescence analysis.

## Neuro-2a culture

Neuro-2a cells (ATCC CCL-131) were maintained in complete medium containing DMEM (Gibco, #11885084), 10% FBS, and 0.4% antibiotic/antimycotic (Gibco, #15240062).

For knockdown studies, a reverse transfection approach was used. A mix of 500 ml/well of OPTI-MEM (Gibco, 31985062), 5 ml DharmaFECT 1.0 (Dharmacon, #T-2001-02), and 25 nM siRNA was prepared and applied to a 6-well plate. Cells were dissociated by trypsinization and prepared as suspension of 350,000 cells/ml in complete medium without antibiotic/antimycotic. Cells were then added to the well and incubated at 37 °C before analysis by qRT-PCR. For Bmal1 overexpression, 350,000 cells were transfected with Bmal1 OE lentivirus (VectorBuilder #VB221206-1193ges, encoding Bmal1 NM_001411976.1 transcript) or pEGIP control lentivirus (Addgene #26777) as previously described[11]. For neurite outgrowth analysis, cells were replated on laminin/PLO coated coverslips in differentiation media containing serum-free DMEM supplemented with 0.5 mM dibutyryl cyclic-AMP (Sigma, D0260). ON-TARGETplus SMART pool siRNA oligos were ordered from Dharmacon (siBmal1, L-040483-01-0005, and non-targeting pool, D-001810-10-05).

## Generation of induced neurons and replating assay

H9 human embryonic stem (hES) cells were obtained from the stem cell core at Mount Sinai, maintained and passaged in mTeSR1 media (Stem Cell Technologies, #85850) using aggregate passaging with ReLeSR (Stem Cell Technologies). 0.4% of antibiotic-antimycotic agent was added to all media used in these studies (Gibco, 15240062). The study was approved by the Embryonic Stem Cell Research Oversight Committee (ESCRO) at Icahn School of Medicine at Mount Sinai.

Induction of neuroprogenitor cells (NPCs) was started on day 4-5 after passaging by dual-Smad inhibition[112], using STEMDiff SMADi neural induction kit (Stem Cell Technologies, #08581). Stem cell colonies were dissociated with Gentle Cell Dissociation Reagent (Stem Cell Technologies, #07174) and resuspended to a final concentration of 3 × 10^6 cells/ml in STEMDiff SMADi media supplemented with 2.5 mM thiazovivin (Calbiochem, #420220). One ml of cell suspension was applied to one well of a preprepared AggreWell 800 plate (Stem Cell Technologies, 34850), and cultured with daily partial (3/4) media changes using STEMDiff SMADi complete media (without thiazovivin) for 4 days. On day 5, EBs were replated on matrigel (Corning, #356238) coated 6-well plate and full media change was performed daily. On day 12, neural rosette selection was performed using STEMdiff Neural Rosette Selection Reagent (Stem Cell Technologies, #05832) and replated in matrigel-coated 6-well plate containing 2 ml of STEMDiff SMADi complete media with daily full media changes until 80% confluence. NPCs were then expanded by passaging with accutase on matrigel-coated 6-well plate at a density of 1.2 × 10^6 in 2 ml in STEMdiff Neural Progenitor Medium (Stem Cell Technologies, #05833). NPCs were characterized by immunostaining with anti-Nestin (R&D,

IC1259G, 1:200) and anti-Pax6 (Biolegend, 901303, 1:100), and frozen at a density of 4.0 × 10^6 cells per ml in STEMdiff Neural Progenitor Medium including 10% DMSO.

Induction of NPCs into mixed cortical culture was performed as previously described[113]. Briefly, NPCs were seeded in a 6-well plate at a density of 360,000 cell/well in BrainPhys media (StemCell Technologies, #05790) supplemented with 1x N2 (Gibco, #17502048), 1× B27 (Invitrogen, 12587-010), 20 ng/ml brain-derived neurotrophic factor (BDNF, Peprotech, #450-02), 20 ng/ml glia-derived neurotrophic factor (GDNF, Peprotech, #450-10), 250 μg/ml dibutyryl cyclic AMP sodium salt (db-cAMP, Stem Cell Technologies, #73884) and 200 μM L-ascorbic acid (Stem cell technologies, #72132). Media changes were performed every other day until indicated. A replating assay was performed as previously described[114]. Briefly, cells were washed twice with PBS and incubated in 0.025% trypsin for 5 min at 37 °C. While cells were still attached, trypsin was gentle removed and differentiation media was added. Cells were subsequently dissociated by pipetting, counted, and seeded in 4-well chamber slides at a density of 55,000 cells/well in differentiation media containing circadian drugs, KL001 (Cayman, #13878) or PF-670462 (Cayman, #14588) at indicated concentrations.

## Immunofluorescence imaging

For immunofluorescence cytochemistry, cultured DRG neurons or induced neurons derived from hES cells were washed once with PBS and fixed in 4% paraformaldehyde containing 4% sucrose in PBS for 15 min at room temperature. NPC fixation was performed in 4% formalin in PBS for 10 min at room temperature. After three 5 min washes with PBS, cells were incubated in blocking buffering (5% donkey serum, 0.3% Triton X-100 in PBS) for 45 min. Primary antibody incubation was performed overnight at 4 °C in antibody dilution buffer (1% BSA, 0.3% Triton X-100 in PBS), followed by 3×5 min washes with PBS. Secondary antibody incubation was performed together with DAPI (Invitrogen, D1306) at room temperature for 1 h, followed by 3x 5 min washes with PBS. Glass coverslips were mounted with Fluoromount-G (SouthernBiotech).

For immunofluorescence histochemistry, dissected DRGs or sciatic nerves were fixed in 4% paraformaldehyde at 4 °C, cryopreserved in 30% sucrose for 24 h at 4 °C, and embedded in Tissue-plus OCT compound (Fisher Healthcare, 4585). 12 μm thick sections were prepared on a cryostat. For immunofluorescence for the 5hmC mark, DRG sections were permeabilized with 0.5% Triton-X for 30 min, washed twice with PBS for 5 min, and incubated with 100 μg/ml RNase A (Thermo Scientific, EN0531) in PBS at 37 °C. After two 5 min washes in PBS, DRG sections were denatured in 2 N HCl for 30 min and then neutralized with 100 mM Tris pH8.5 for 30 min. Sections were then washed three times for 5 min with PBS, blocked for 1 h, and immunostained as described above.

For wholemount immunofluorescence of paw pads, hind paw skin (ipsilateral to injury) was dissected, cleaned from fat and connective tissue, washed three times for 5 min with PBS, and post-fixed in 4% PFA at 4 °C. Tissue was rinsed thrice for 5 min in PBS and washed ten times for 30 min each with PBS containing 0.3% Triton X-100 (0.3% PBST). Primary antibody incubation was performed in blocking buffer (0.3% PBST containing 5% goat serum and 20% DMSO) at room-temperature for 5 days with gentle shaking. Tissue was washed ten times for 30 min each with 0.3% PBST and incubated with secondary antibodies in blocking buffer at room-temperature for 3 days with gentle shaking. After washing ten times for 30 min with 0.3% PBST, tissue was dehydrated in 50% methanol for 5 min, 100% methanol for 20 min, and finally cleared in benzyl alcohol: benzyl benzoate mix (1:2 ratio) overnight at room temperature. Tissue was imaged in clearance solution with a Zeiss LSM 780 confocal microscope.

Immunofluorescence images were captured with a Zeiss Axio Imager.A2 microscope with AxioCam MRm camera. For β-catenin imaging and whole mount nerve reinnervation imaging, maximal

intensity projections of z-stacks that were captured with a Zeiss LSM 710 confocal microscope were used. Brightfield imaging was performed using an Olympus CKX53 microscope.

Primary antibodies used were as follows: anti- Bmal1 (Novus, NB100-2288, 1:250), TET3 (Active Motif, #61744, 1:300), 5hmC (Active Motif, #39769, 1:500), Tet3 (Active motif, #61743, 1:300), pCreb$^{S133}$(Cell Signaling, #9198, 1:300), β-catenin (BD Biosciences, #610153, 1:200), TUJ1/β3 tubulin (Biolegend, #801202, 1:1000; Cell Signaling, #5568, 1:300), SCG10 (Novus, NBP1-49461, 1:1000), CD206 (R&D systems, AF2535, 1:200), CD68 (Bio-Rad, MCA1957GA, 1:200), IBA1 (Wako, #019-19741, 1:2000), ATF3 (Santa Cruz, sc-188, 1:300), PGP9.5 (Neuromics, RA12103), NFH (EMD Millipore, AB5539, 1:1000), and GFP (AVES, AB5541, 1:1000), NPY (Santa Cruz, sc-133080, 1:100), GAP43 (Millipore, AB5220, 1:300), mCherry (Invitrogen, M11240, 1:500). Antibody validation information including citations are available on the manufacturer's website. Secondary cross-adsorbed and Alexa-coupled donkey anti-IgG antibodies (Jackson Immunoresearch) were used at a dilution of 1:300.

### RNA isolation and qRT-PCR analysis

DRGs were isolated and immediately placed in RNAlater stabilizing solution (Thermo Scientific, AM7024). Total RNA isolation was performed using the RNeasy plus micro kit (Qiagen, 74034). Briefly, DRGs were completely lysed in 80 μl of RLT buffer supplemented with β-mercaptoethanol using RNase-free pestles (Fisher, #12-141-368). Subsequently, additional RLT buffer was added to a final volume of 350 μl and samples were vigorously pipetted before proceeding with RNA isolation. RNA isolation from H9 hES cells, NPCs, and neurons was performed using standard TRIzol extraction (Invitrogen, #15596026). Reverse transcription was performed using Superscript III reverse transcriptase (Invitrogen, #18080051). Real-time qRT-PCR was performed using ABI Prism 7900HT Sequence Detection instrument and SYBR green master mix (QuantaBio, #95073-012). Gene specific primers (Supplementary Data 14) were purchased from Integrated DNA Technologies. Melting curves were performed for each run to ensure specificity of the product and the housekeeping genes *Hprt* and *GAPDH* were used for normalization of mouse and human samples, respectively. Relative gene expression was analyzed according to the ΔΔCt method. For conditioned DRG analysis, ipsilateral L4–L6 DRGs were pooled from 2 mice for each sample.

### Co-immunoprecipitation assay and Western blot analysis

Co-immunoprecipitation was performed based on protocols from the Longworth laboratory[115]. Neuro-2a cells were cultured in low glucose DMEM (Gibco, 11885084) containing 10% FBS. On the first day, Neuro-2a cells were seeded at a density of $8.5 \times 10^6$ cells in a T75 flask for each immunoprecipitation sample. On day3, cells were washed once with PBS and serum-starved for 24 hrs. For each sample, lysis was performed in 1 ml of ice-cold high salt lysis buffer containing 300 mM NaCl, 50 mM Tris pH 7.5, 1 mM EDTA, 0.1% Triton, 10% glycerol, 1 mM DTT, 1× complete mini protease inhibitors (Sigma, 11836170001) for 1 h at 4 °C. Lysate was cleared by centrifugation at 13,000 g for 15 min at 4 °C. 50 ml of protein G Dynabeads (Thermo scientific, 10003D) were used for each immunoprecipitation sample. Beads were washed two times with blocking buffer (0.5% BSA, 0.2% sodium azide in PBS) and blocked for 30 min at room-temperature under shaking. Anti-Tet3 antibody (Active motif, #61743, 1:100) or IgG control (Millipore, NI04-100UG) was conjugated to the beads in blocking buffer for 4 h at room-temperature. Beads were washed twice with blocking buffer and lysate was applied and incubated overnight at 4 °C. Next day, beads were washed four times 5 min each with ice-cold 1× PBS at 4 °C, then suspended in 60 ml of 2× NuPAGE LDS buffer (Invitrogen, NP0007) with β-mercaptoethanol. For DRG analysis, extraction was performed in RIPA buffer containing 1× complete mini protease inhibitors (Sigma, 11836170001) and 1× PhosSTOP phosphatase inhibitor (Sigma,

4906845001) by homogenization on ice with pestles and rotating at 4 °C for 30 min. Subsequently, lysate was centrifuged at $13,000 \times g$ for 10 min at 4 °C, boiled in 1× NuPAGE LDS sample buffer, and used for analysis. Western blot detection was performed using anti-Tet3 (Active motif, #61743, 1:1,000), anti-Bmal1 (Novus, NB100-2288, 1:1000), or anti-β-actin (Sigma, A1978, 1:10,000) antibodies as previously described[116] and membranes were scanned with LI-COR Odyssey Classic system.

### Image analysis

Image analysis was performed with the Fiji package of ImageJ[117]. Prior to importing the Tiff image into ImageJ, the global scale was set according to the magnification used. For measurement of neurite length, a simple neurite tracer (SNT) tool was used to determine the length of longest neurite for each DRG neuron[118]. Explant cultures were analyzed with Neurite J 1.1 plugin as previously described[119]. For analysis of axon regrowth in injured nerve, micrographs of nerves were stitched together using Microsoft Paint. Images were then imported into ImageJ and a rectangle was drawn to outline the length of the nerve. The SCG10 immunointensity profile along the length of the nerve was determined using the Plot Profile tool. Finally, the intensity profile plot values were normalized to the highest intensity value, representing the beginning of the crush site.

Micrographs of footpads were analyzed with maximum intensity projections of z-stack images. PGP9.5 or NFH positive areas were measured by thresholding and using the Analyze Particle (with limit to threshold) summary tool. Total footpad area was measured by manual tracing and using the 'Measurement' tool.

5hmC, Tet3, pCREB$^{S133}$, and ATF3 immunofluorescence intensities in DRG neurons were analyzed by thresholding and using the Analyze Particle tool by setting the minimal particle size limit to 50 μm and the circularity value to 0.8-1 to exclude staining of the smaller satellite cells. Quantification of the expression of macrophage markers was performed on two randomly selected 100 mm$^2$ areas of DRG images, using the Analyze Particle tool. For all analyses, thresholding values were kept constant between groups. Analysis of β-catenin localization was performed by measuring the average IF intensity of randomly selected areas of membrane or cytoplasm and calculating the ratio of membrane to cytoplasm localization.

### RNA-seq analysis

RNA samples of DRG tissue and dissociated neurons were collected with the QIAGEN RNeasy plus mini kit and sent to Psomagen, Inc for next-generation sequencing on the Illumina NovaSeq platform. The NGS-Data-Charmer pipeline was used for preprocessing, QC, and alignment of the FASTQ files (https://github.com/shenlab-sinai/NGS-Data-Charmer). In brief, adapter trimming was achieved by using the Trim-Galore tool[120] (v0.6.5), followed by alignment of trimmed reads to the mouse mm10 genome assembly using Bowtie2[121] (v2.4.1), and duplicated read pairs were further removed using the 'rmdup' module of SAMtools[122] (v1.10). featureCounts was used to obtain a gene expression matrix, using the parameters "--fraction -t gene" on the GENCODE annotation (vM25). Genes were filtered to remove lowly expressed genes, defined as fewer than 5 samples showing a minimum read count of 2 reads, prior to performing differential analysis with DESeq2.

### Bioinformatics

Unbiased transcription factor (TF) binding motif enrichment analysis of DhMRs from DRGs after sciatic nerve lesion (previously reported in our study[17]) was performed with the Homer package[123], using the Perl script findMotifsGenome.pl. Background sequences were chosen at random from the mouse mm9 genome. Targeted enrichment analysis of motifs was performed with the Perl script annotatePeaks.pl. Chromatin accessibility of DhMRs and associated genes after nerve injury

(using ATAC-seq data from study[14]) was determined by deepTools software[124]. Regeneration associated genes (RAGs) identified by bulk RNA-seq from whole DRGs were obtained from data of previously published RNA-seq studies using a cutoff criterion of FDR < 0.2[14,85]. Genes that are associated with DhMRs are determined using the Genomic Regions Enrichment of Annotations Tool (GREAT)[35], using the mouse mm9 genome build and whole genome background and default settings for analysis, yielding approximately two associated genes per DhMR. Pathway enrichment analysis of gene sets was performed with the Enrichr platform[125], the Gene Ontology platform[126], and Ingenuity Pathway Analysis (Qiagen)[127]. Tet3 promoter analysis was performed using the Eukaryotic Promoter Database platform[128].

### Statistical analysis

All statistical analyses were performed using GraphPad Prism 9. For comparison between two groups, unpaired two-tailed t-test was performed on samples with normal distribution, while Mann-Whitney test was applied to samples that did not display normal distribution. Kruskal-Wallis test with Dunn's multiple comparisons test was used for multigroup comparison. For repeated measures, Two-way ANOVA with Bonferroni's multiple comparison test was performed. Bar graphs in all figures represent mean ± SEM. Differences were considered as significant with P values *P < 0.05, **P < 0.01, ***P < 0.001.

### Reporting summary

Further information on research design is available in the Nature Portfolio Reporting Summary linked to this article.

## Data availability

Source data are provided with this paper. The RNA-seq data has been deposited at the NCBI GEO database under accession number GSE233367. The DRG 5hmC sequencing dataset[17] is available under accession code GSE85972. Published RNA-seq and ATAC-seq datasets[14] are available under accession codes GSE97090 and GSE132382. Source data are provided with this paper.

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

## Acknowledgements

We thank Chrystian Junqueira Alves and Sangjo Kang from the Zou laboratory for assistance with confocal imaging, Nada Marjanovic and the Mount Sinai Genomics Core for assistance with qRT-PCR, Julia TCW for feedback on neuronal differentiation techniques, Gill Bejerano and Yosuke Tanigawa for advice on GREAT analysis, Erik Musiek for advice on chronobiology, and all members of the Zou and Friedel laboratories for helpful discussions. This work was supported by funds to H.Z. from NIH (R01 NS127442), the New York State Spinal Cord Injury Research Board (DOH01-C33268GG, DOH01-C30832GG, and DOH01-C32242GG), and Neilsen Foundation (#890112). Y.W. was partly supported by scholarship fund from Xi'an Jiaotong University.

## Author contributions

Conceptualization: D.H., X.H., R.H.F., H.Z.; Data acquisition: D.H., Y.W.; Data analysis: D.H., Y.W., R.H.F., H.Z.; Bioinformatic analysis: A.R., D.H., M.E., L.S.; Manuscript writing: D.H., R.H.F., H.Z.

## Competing interests

The authors declare no competing interests.
