## [Peer Review file · Nature Communications]

REVIEWER COMMENTS

Reviewer #1 (Remarks to the Author):

Successful axon regeneration requires epigenetic reconfigurations (DNA and histones) to rewire the gene regulatory circuits, and in turn establishes regenerative program to increase growth capacity. DNA methylation is one of epigenetic barriers confining neurons at the dormant regenerative state. Naturally, peripheral lesions can induce TET3 expression to reprogram epigenetic states in favoring axon regeneration. Perturbations of TET3 resulted in impaired axon regrowth and function recovery. While these results from the previous studies have suggested that the necessity of TET3 or DNA demethylation in the regeneration processes, whether the augments of TET3 (or the resulted 5hmC changes) play an instructive role in promoting regenerative capacity has never been addressed before.

In this study, the authors identified Bmal1 as a new regulator of TET3 and examined the role Bmal1 in axon regeneration. The authors previously identified ~1000 DhMRs in DRGs upon peripheral lesion. They reanalyzed the sequencing result and searched for statistically significant over-represented TF binding motifs as some of these TFs may collaborate with Tet3 to regulate intrinsic growth capacity. Intriguingly, the analysis indicated an enrichment binding of Bmal1, which is a central component of the circadian clock. They then applied conditional gene deletion in conjunction with in vitro and in vivo models where the results indicated that neuronal Bmal1 deletion enhanced axon regeneration via Tet3-dependent manner. Further, they showed diurnal oscillation of TET3/5hmC in DRG neurons, which is anti-phasic to Bmal1 transcriptional activity.

Overall, their work supported their conclusions and claims. This is an exciting study that they identified Bmal1 as an approach/pathway/target to module TET3 level and demonstrated the TET3 (or 5hmC) augments can promote axon regeneration. Their work advance our knowledge of epigenetic regulations of regenerative capacity. I recommend this study to be published in Nature Communications if the following concerns can be addressed.

Major/ Minor concern:

1. In Fig 1B, the authors performed TF analysis on PL-specific DhMRs and identified Bmal1 as the top candidate. Do these DhMRs for TF analysis include both gain and loss of 5hmC regions? Will these two catalogs of DhMRs yield different TF enrichment?
2. In Fig 1D. Co-IP performed in Neuro-2a cell suggests the physical interaction between Tet3 and Bmal1. It will be more relevant if the authors could show the Tet3 and Bmal1 interactions in DRGs using proximity ligation assay.
3. Fig. 2C and 2D. The figures are incorrectly placed. Need to swap.

Fig2C should the time course analysis. Fig.2D should the immunostaining (Line 140-141).

4. In Fig. 2C (original). The neurons sizes are relatively large in the represented images of PL (1 dpi). The authors may consider retaking a better image.

5. The author mentioned that the Bmal1 protein level was not changed after PL using WB assay and that was due to the substantial proportion of glial cells in the tissue. Alternatively, this could result from only certain types of neurons having Bmal1 reduction. The authors may utilize publicly available single-cell RNA seq data to examine the Bmal1 expression was only changed in certain neuron populations.

6. Fig. 5A; Line 263-264. "To further explore the collective contribution of Bmal1-Tet3-5hmC regulon to axon regeneration, we combined the 251 RAGs (with 14 redundant genes from the two sets) for integrated analysis (Fig. 5A). The RAGs overlapping with DhMRs (containing Bmal1 motif) will make more sense for this analysis. If the authors focus the Tet3-5hmC regulon (which is the title of Figure 5), the combined 251 RAGs are rationally selected.

7. Line 1252: should be nuclear protein levels; not clear protein levels.

Reviewer #2 (Remarks to the Author):

The role of circadian rhythms in tissue regeneration is a relatively unexplored field. In this paper, the authors discover there is Bmal1 activity during peripheral axon regeneration. They propose that Bmal1 regulates DNA demethylation by negatively regulating the epigenetic factor Tet during axon regeneration. The paper is well-written and the data are of good quality in most cases, data showing Bmal1 regulation of Tet3 are the most convincing. The demonstration that neurite growth and nerve regeneration changes by time of day (Fig 7) is exciting but then the data showing the same occurs in the Bmal1 cKO raises questions about whether this is circadian-regulated. Additionally, the authors only test one or two time points throughout the study, this also raises questions about the circadian nature of axon regeneration (see comments below).

The study adds to a growing body of work that shows different roles of circadian clock components in tissue regeneration. The finding that clock components participate in neuron regeneration is new, and

the demonstration that Bmal1 is a negative regulator of transcription via Tet3 is important. The study would be well-received in the field given several points are addressed:

1. Fig S2, the authors perform assays to show Bmal1 loss leads to increased neurite growth. Would Bmal1 overexpression in Neuro2a cells or ES-derived cells reduce growth?

2. Fig 2C needs to more clearly show TuJ1+ cells, and show the reduction in Bmal1 expression specifically in neurons during damage. In this image, and others in the paper it is not clear which cells the authors are pointing out. These should be outlined or arrows added to the images to clarify. Is the Bmal1 antibody working? It is not clear if Bmal1 is cycling, or at what time was the sample obtained. Similar images in Fig S3E-F (which are actually better quality) also do not clearly state time of collection.

3. Fig 4A-B (and Fig S5), the quantification and images do not seem to correspond. The images show a drastic difference that is much lower in the quantifications, better representative images should be shown in these and other figures.

4. Results pg 11-12 “Bmal1-Tet3-5hmC epigenetics impact chromatin accessibility...” referring to Fig 5C are confusing. The b-catenin levels are much lower in the cKO cells, what little remains is at the membrane. However, Fig S7D shows that Wnt pathway components are upregulated, how would this work if b-catenin is nearly absent in the cKO nuclei? Are the gene analyses in Fig S7-8 showing upregulated genes in the Bmal1 cKO mutant, or are they showing genes positively regulated by Bmal1? Clarification is needed to understand this section.

5. Fig 6: The authors conclude that “Hence ZT0-3 and ZT12-15 correspond to opposite phases of DRG circadian clock.” (p 14). Opposite phases in circadian biology means the peak and troughs of gene expression are anti-phasic in the context of cyclic expression over 24 hour time. Conclusions about circadian or diurnal expression in this figure, and others, are too strong given that only two time points over 24 hours were tested. First, it is not clear that these represent either the peak or trough of Bmal1 and Tet3 expression. Second, it is not clear if these genes are functionally correlated, Tet3 might be changing due to the light-dark cycle, while Bmal1 is circadian, and actually both are not opposite to each other at all. Third, sampling only two times can miss rhythms. As shown in 6A the circadian clock gene Nr1d1 is not even rhythmic, nor are most of the genes tested in Fig S10. When the authors later state: “Having demonstrated diurnal oscillation of Tet3-5hmC that is anti-phasic to Bmal1 transcriptional activity in DRG neurons” (p 16), these statements ignore limitations of the data and are far too strong. This same problem also applies to Fig S11E where the authors claim genes show “transcriptional oscillation” based on only two timepoints tested. If the authors want to conclude rhythms are present, they need to examine many more time points over 24 hours. At least the model proposed in Fig 6E should be tested by examining Bmal1 protein by antibody at more than one time point to show that it is higher at ZT0-3.

6. Fig 8: The authors show that the Bmal1 cKO has the same daily change in neurite growth and regeneration as the control. This data suggests Bmal1 regulates Tet3 non-rhythmically, acting as a negative regulator but not necessarily in a circadian manner. The data showing that Bmal1 decreases during injury (Fig 2C-D), further suggests that Bmal1 cannot be responsible for the time of day changes in neuron regeneration – it is absent to play a role during the stress response. These data challenge the schematic shown in Fig 8H. Another factor must be responsible, maybe light-response, sleep, or other neuronal activity that coordinate with Bmal1 to effect these changes. The authors should consider these. Alternatively, it is possible the authors have missed the circadian interaction by only testing at two times of day. What are Tet3 expression and 5hmC levels in the Bmal1 cKO at these and different times? Sampling at only the morning and evening may have missed a stronger interaction between Bmal1 and Tet3/regeneration, a maximum or minimum that is not ZT0-3 or ZT12-15 that would better support the model the authors propose.

7. Fig S12: The authors show that Creb is higher in the evening (A-B) then show that Creb activation in Bmal1 cKO through cAMP does not further increase neuron growth. Creb should be tested in Bmal1 cKO.

8. Statistics: The authors should provide a supplementary table showing full results of the statistics throughout the paper. For example in Fig 8D, a two-way ANOVA and Bonferroni is done but only the p-value is given.

Reviewer #3 (Remarks to the Author):

Halawani et al. describe a new link between the circadian transcriptional regulator BMAL1 and the Tet3 DNA methylcytosine dioxygenase in peripheral axon regeneration. They also report diurnal oscillations in DNA hydroxymethylation that may contribute to differences in axonal growth throughout the day. Together, these are interesting observations, but the key findings are not sufficiently nailed down. This limits unfortunately limited my enthusiasm for the manuscript.

Major:

A key claim of this paper is that Bmal1 and Tet3 directly interact, but the only data provided to support this is a one-way IP a cell line. I recommend the authors consider IPing with Bmal1 and seeing if Tet3 comes down as well as investigate whether this interaction occurs in DRG.

While the CreER system is very powerful, it is important to confirm the the core findings are not secondary to tamoxifen itself. I believe this control is missing for the axon outgrowth, Tet3 expression levels, macrophage activation, etc. in the Bmal1 cKO.

The evidence to support the Bmal1 cKO phenotypes are through Tet3 is not strong. There is a very subtle phenotype with the Tet3 inhibitor C35, but this would be stronger with a dose response over wider range as well as some genetic evidence as these tools are widely available.

Additional time points would be helpful throughout the day would be helpful for understanding the circadian nature of gene expression, DNA hmC and axonal growth.

Bmal1 cKO does not seem sufficient to block the diurnal effects on axonal growth, what factors are driving this? Do Clock, Per KOs display related axonal growth phenotypes after injury?

Are there diurnal changes in RAG induction and 5hmC that are affected by Bmal1 cKO? 5hmC immunofluorescence is rather non-specific. Locus specific measurements would be more compelling and mechanistically interpretable.

Minor

Is the difference in Fig 1F statistically significant? Would increased ATAC signal be seen at any gene that is unregulated after PL regardless of its hydroxymethylation status?

How specific is the Bmal1 antibody for the IF staining in Fig 2C? This quantification would be more convincing if staining from the Bmal1 f/f cKO were shown.

Fig S3E: shouldn't Bmal1 protein only be downregulated in Thy1-positive neurons? These are not displayed in the image, but it appears that all Bmal1 staining is weaker in the example. Co-staining with Cre could help clarify this.

Changes in pCREB and b-Catenin levels in Bmal1 cKO mice are not specific to the enriched bioinformatic pathways in Fig 5. It might be more relevant to use inhibitors of various arms of the enriched pathways to see if this affects the downstream up regulation of pCREB or b-catenin.

Recommend copy editing of manuscript as there are a number of grammatical errors and typos

RESPONSE LETTER

Point-by-point response:

Reviewer #1:

Successful axon regeneration requires epigenetic reconfigurations (DNA and histones) to rewire the gene regulatory circuits, and in turn establishes regenerative program to increase growth capacity. DNA methylation is one of epigenetic barriers confining neurons at the dormant regenerative state. Naturally, peripheral lesions can induce TET3 expression to reprogram epigenetic states in favoring axon regeneration. Perturbations of TET3 resulted in impaired axon regrowth and function recovery. While these results from the previous studies have suggested that the necessity of TET3 or DNA demethylation in the regeneration processes, whether the augments of TET3 (or the resulted 5hmC changes) play an instructive role in promoting regenerative capacity has never been addressed before.

In this study, the authors identified Bmal1 as a new regulator of TET3 and examined the role Bmal1 in axon regeneration. The authors previously identified ~1000 DhMRs in DRGs upon peripheral lesion. They reanalyzed the sequencing result and searched for statistically significant over-represented TF binding motifs as some of these TFs may collaborate with Tet3 to regulate intrinsic growth capacity. Intriguingly, the analysis indicated an enrichment binding of Bmal1, which is a central component of the circadian clock. They then applied conditional gene deletion in conjunction with in vitro and in vivo models where the results indicated that neuronal Bmal1 deletion enhanced axon regeneration via Tet3-dependent manner. Further, they showed diurnal oscillation of TET3/5hmC in DRG neurons, which is anti-phasic to Bmal1 transcriptional activity.

Overall, their work supported their conclusions and claims. This is an exciting study that they identified Bmal1 as an approach/pathway/target to modulate TET3 level and demonstrated the TET3 (or 5hmC) augments can promote axon regeneration. Their work advance our knowledge of epigenetic regulations of regenerative capacity. I recommend this study to be published in Nature Communications if the following concerns can be addressed.

>> We thank the Reviewer for positive comments. We have conducted a series of new studies to address the critiques.

Major/ Minor concern:

RI.1. *In Fig 1B, the authors performed TF analysis on PL-specific DhMRs and identified Bmal1 as the top candidate. Do these DhMRs for TF analysis include both gain and loss of 5hmC regions? Will these two catalogs of DhMRs yield different TF enrichment?*

>> Done. The initial study was conducted on PL-DhMRs with 5hmC gain or loss combined. We have since re-analyzed the PL-specific DhMRs, separating into 5hmC gain (54%) or loss (46%). We found that these two catalogs of PL-DhMRs yielded shared (e.g., ARNT, NPAS1), but also distinct TF enrichment (new **Fig. S1b**; new **Table S1**). Notably, the bHLH-PAS family was featured in both groups, but more prominently for the DhMRs with 5hmC gain (e.g., CLOCK, HIF-1A, bHELHE40, NPAS2), while GATA binding motifs were featured for the 5hmC loss DhMRs.

Throughout the manuscript, whenever possible, we also analyzed the two groups of PL-DhMRs separately, including chromatin accessibility and associated genes. Results showed that both catalogs of PL-DhMRs were predominantly localized in gene body and intergenic regions, rather than promoters and CpG islands (new **Fig. 1b**). ATAC-seq data showed that PL led to increased chromatin accessibility for both DhMRs, but more so for group with 5hmC gain, while the associated genes for both groups showed more open chromatin at the transcription start site (TSS) after PL (new **Fig. S1d**). For the associated genes, the ones with 5hmC gain (n=988) concerned epithelial cell proliferation and differentiation, organ regeneration, and lymphocyte co-stimulation, while the genes with 5hmC loss (n=834) mainly concerned metabolism (e.g., response to thyroid hormone, amino acid biosynthetic, glucose homeostasis) and cytoskeletal reorganization (new **Fig. S1d**; **Table S2**). In this context, it is noteworthy that as an intermediate along the DNA demethylation pathway, 5hmC gains or losses do not strictly correlate with transcriptional gain or loss.

RI.2. In Fig 1D. Co-IP performed in Neuro-2a cell suggests the physical interaction between Tet3 and Bmal1. It will be more relevant if the authors could show the Tet3 and Bmal1 interactions in DRGs using proximity ligation assay.

>> We have performed new Co-IP on DRG lysates, showing that IP with a specific antibody against Tet3¹⁻³ could pull down Tet3 along with Bmal1 (new **Fig. S2a**). Interestingly, unlike in Neuro-2a lysate with a predominant band of Tet3 at ~185 kDa, we observed a dominant band of Tet3 at ~80 kDa in DRG lysates, while pull down of Tet3 resulted in a faint band at ~185 kDa and multiple bands of lower molecular weight (new **Fig. S2a**), in agreement with the report that Tet3 can be proteolytically processed by calpain cleavage, producing ~80 kDa band⁴. Since the anti-Tet3 antibody binds to the C-terminal catalytic domain, these fragments are consistent with N-terminal proteolysis, a susceptible region with low structural complexity (new **Fig. S2b**). Of note, Tet3 is also a substrate of O-linked-N-acetylglucosamine (O-GlcNAc) transferase², and we found that removal of O-GlcNAc by β -

Figure. Tet3 modification by O-GlcNAc transferase stabilizes Tet3 in Neuro2a lysates, as removal of O-GlcNAc led to disappearance of the band of Tet3 at 80 kDa.

hexosaminidase in Neuro-2a lysate led to disappearance of the band at ~80 kDa (see Figure on right).

We also attempted reverse co-IP with a specific antibody against the C-terminal region of Bmal1 (validated with *Bmal1* knockout cells) but failed to pull down either endogenous or overexpressed Tet3 in Neuro-2a cells (new **Fig. S2c**). This may be related to a low abundance of endogenous Tet3 or binding of Tet3 to the C-terminal domain of Bmal1 (thus not accessible for anti-Bmal1 antibody). To address the former possibility, we overexpressed the short or long isoforms of Tet3 in Neuro-2a cells, but still failed to co-IP Tet3 with Bmal1 antibody (new **Fig. S2c**). Indeed, the C-terminal domain of Bmal1 contains a conserved transactivation domain prominently involved in binding to transcriptional regulators^{5,6}. We looked into proximity ligation assay, but the kits were not compatible with rat anti-Tet3 antibody (validated here) and we have tested a number of antibodies (rabbit, mouse, or goat) for mouse Tet3 using knockdown or CRISPR-knockout cells as negative controls, but did not find specific antibodies.

RI.3. Fig. 2C and 2D. The figures are incorrectly placed. Need to swap. Fig2C should the time course analysis. Fig.2D should the immunostaining (Line 140-141).

>> Corrected. We appreciate the comment. They are now **Fig. 2d** and **2e**.

RI.4. In Fig. 2C (original). The neurons sizes are relatively large in the represented images of PL (1 dpi). The authors may consider retaking a better image.

>> Done. We provided more representative images (new **Fig. 2e**), with arrows pointing at examples of DRG neurons for clarity.

RI.5. The author mentioned that the Bmal1 protein level was not changed after PL using WB assay and that was due to the substantial proportion of glial cells in the tissue. Alternatively, this could result from only certain types of neurons having Bmal1 reduction. The authors may utilize publicly available single-cell RNA seq data to examine the Bmal1 expression was only changed in certain neuron populations.

>> Done. We appreciate the suggestion. We surveyed a recent single cell RNA-seq dataset⁷, which revealed that 8 out of 9 subclusters of DRG neurons downregulated *Arntl* (*Bmal1*) at 24 hrs post-PL, and this coincided with *Tet3* upregulation in the same DRG neurons (new **Fig. S3a**).

RI.6. Fig. 5A; Line 263-264. "To further explore the collective contribution of Bmal1-Tet3-5hmC regulon to axon regeneration, we combined the 251 RAGs (with 14 redundant genes from the two sets) for integrated analysis (Fig. 5A). The RAGs overlapping with DhMRs (containing Bmal1 motif) will make more sense for this

analysis. If the authors focus the Tet3-5hmC regulon (which is the title of Figure 5), the combined 251 RAGs are rationally selected.

>> Corrected. We appreciated the careful review. We renamed it as Tet3-5hmC regulon (new **Fig. 6a-c**). Of note, we replaced the RAGs with our own new RNA-seq data on axotomized DRG at 1 dpi after PL vs. no injury, *with defined ZT2-4*. This resulted in 346 genes, rather than the 251 genes using published RNA-seq data from unspecified ZT.

RI.7. Line 1252: should be nuclear protein levels; not clear protein levels.

>> Corrected.

Reviewer #2:

The role of circadian rhythms in tissue regeneration is a relatively unexplored field. In this paper, the authors discover there is Bmal1 activity during peripheral axon regeneration. They propose that Bmal1 regulates DNA demethylation by negatively regulating the epigenetic factor Tet during axon regeneration. The paper is well-written and the data are of good quality in most cases, data showing Bmal1 regulation of Tet3 are the most convincing. The demonstration that neurite growth and nerve regeneration changes by time of day (Fig 7) is exciting but then the data showing the same occurs in the Bmal1 cKO raises questions about whether this is circadian-regulated. Additionally, the authors only test one or two time points throughout the study, this also raises questions about the circadian nature of axon regeneration (see comments below).

>> We are grateful for the thoughtful review and positive comments.

We suspect that the reason that DRG neurons from *Bmal1^{cKO}* mice (maintained in regular 12 hr light:dark cycle) continued to display time-of-day effect is related “masking”, a response to light overriding the absence of circadian clock⁸⁻¹⁰. Light is the strongest *zeitgeber* (“time giver”) that entrains the central clock located in the suprachiasmatic nucleus of the mammalian brain¹¹.

We have conducted new mapping of expression changes of Bmal1 and Tet3-5hmC in DRG neurons every 4 hrs along ZT, and neurite outgrowth capacity every 6 hrs (see below). We agree with the reviewer that future studies are needed to determine whether Tet3-5hmC oscillation and time-of-day effect on axon growth potential are truly circadian (controlled by Bmal1 clock), or related to *light cycle, sleep, or neuronal activity*. We added in Discussion that to distinguish circadian control (mediated by Bmal1) vs. light entrainment of diurnal changes of Tet3/5hmC and neurite outgrowth in DRG, future studies are needed using mice maintained in constant darkness (DD). If axon growth capacity is truly circadian, DRG neurons from mice kept in DD should still show rhythmic changes of neurite outgrowth. If it is mainly driven by light entrainment (which can affect neuronal

activity and behavioral rhythm), then neurite outgrowth capacity of DRG neurons from mice kept in DD will become arrhythmic.

Future study (outside the scope of this manuscript) can also test phase shift, wherein the light-dark cycle is suddenly shifted by 6 or 12 hrs. Since it takes the circadian system a few days to adjust to the new light cycle (akin to jetlag), the circadian clock will be initially out of phase with the light cycle. This will allow one to tease apart if circadian clock or light entrainment plays a dominant role here. If it is truly circadian, neurite outgrowth will still peak at the old ZT initially, but then gradually shift towards the new timeframe as the clock retrain. In summary, our current work uncovers a novel time-of-day impact on Tet3-5hmC epigenetics and axonal growth, thus providing a starting point for future in-depth studies in this direction.

The study adds to a growing body of work that shows different roles of circadian clock components in tissue regeneration. The finding that clock components participate in neuron regeneration is new, and the demonstration that Bmal1 is a negative regulator of transcription via Tet3 is important. The study would be well-received in the field given several points are addressed:

>>We appreciate the recognition the novelty of our work.

***R2.1.** Fig S2, the authors perform assays to show Bmal1 loss leads to increased neurite growth. Would Bmal1 overexpression in Neuro2a cells or ES-derived cells reduce growth?*

>> Done. Following the advice, we have performed new gain-of-function studies, showing that forced Bmal1 overexpression in Neuro-2a cells attenuated neurite outgrowth (new **Fig. 3c**). Notably, unlike in Neuro-2a cells, over-expression of Bmal1 in primary DRG neurons (conditioned during the dissociation) did not cause significant reduction of neurite outgrowth (new **Fig. S6c**), suggesting that the conditioning effect can bypass or overcome the Bmal1 block.

***R2.2.** Fig 2C needs to more clearly show TuJ1+ cells, and show the reduction in Bmal1 expression specifically in neurons during damage. In this image, and others in the paper it is not clear which cells the authors are pointing out. These should be outlined or arrows added to the images to clarify. Is the Bmal1 antibody working? It is not clear if Bmal1 is cycling, or at what time was the sample obtained. Similar images in Fig S3E-F (which are actually better quality) also do not clearly state time of collection.*

>> Done. We replaced Fig. 2C with more representative images with arrowheads clearly labeling DRG neurons, as well as time of collection (new **Fig. 2e**). We also added time of collection throughout the manuscript and Figures wherever such information is available. Of note, our first part of studies (Figs 1-7) was conducted

without the clear knowledge of the time-of-day effect (a main discovery shown later in Fig. 8-10), thus some earlier studies used mixed ZTs, but main experiments have been repeated at different ZT times.

To our knowledge, we have used the best characterized Bmal1 antibody in the circadian field (Novus 100-2288) that is routinely used by the Takahashi and Musiek groups. Additionally, we have validated the specificity of the antibody showing Bmal1 ablation in DRG neurons of cKO mice by IF staining (**Fig. S5e-f**). As mentioned to Reviewer 1, it was difficult to capture Bmal1 reduction in *Bmal1*^{cKO} DRG or Bmal1 oscillation by WB using DRG tissue lysates (**Fig. S3b**, new **Fig. S14c-d**), perhaps related to a large quantity of glial cells (which may have different cycling phase). Notably, WB did reveal a slight shift of Bmal1 to higher molecular weight after PL (**Fig. S3b**), or at ZT12-15 relative to ZT0-3 (**Fig. S14c**), which may reflect post-translational modifications such as phosphorylation^{12, 13}.

R2.3. Fig 4A-B (and Fig S5), the quantification and images do not seem to correspond. The images show a drastic difference that is much lower in the quantifications, better representative images should be shown in these and other figures.

>> Done. We have now provided more representative images to show increased Tet3 levels in Bmal1-deficient DRG neurons (new **Figs. 5a-b, Fig. S7a-b**). We also re-graphed the violin plots by adjusting the y-axis scale, which more clearly showed the changes of mean immunointensity of Tet3 (new **Figs. 5a-b**).

R2.4. Results pg 11-12 “Bmal1-Tet3-5hmC epigenetics impact chromatin accessibility...” referring to Fig 5C are confusing. The b-catenin levels are much lower in the cKO cells, what little remains is at the membrane. However, Fig S7D shows that Wnt pathway components are upregulated, how would this work if b-catenin is nearly absent in the cKO nuclei? Are the gene analyses in Fig S7-8 showing upregulated genes in the Bmal1 cKO mutant, or are they showing genes positively regulated by Bmal1? Clarification is needed to understand this section.

>> We have revised the section on chromatin accessibility and Wnt/β-catenin.

We now provide more representative images and added quantification, showing that the overall level of β-catenin appeared comparable, but the membrane localization appeared higher in *Bmal1*^{cKO} DRG than controls, signifying inactivation of Wnt pathway (new **Fig. 6e**). This is in line with new IPA bioinformatics revealing that Tet3-5hmC regulon in response to PL was enriched for Wnt/β-catenin with a negative activation z-score (new **Fig. 6c**). In echo, IPA upstream regulator analysis predicted TCF4^{14, 15}, also with a negative activation score (new **Fig. 6d**). Of note, the previous Enrichr analysis did not have information regarding activation or downregulation.

As a further validation of *Bmal1* regulated genes, we conducted new RNA-seq on primary DRG neurons from *Bmal1^{CKO}* vs. control mice (new **Fig. 7**). IPA showed that the *Bmal1* regulon (n=625 DEGs, *Bmal1^{CKO}* / control) was enriched for Wnt/ β -catenin pathway, with a negative activation z-score (new **Fig. 7d**). Similar findings also held true for the *Bmal1* regulon containing PL-DhMR (new **Fig. 7f**). Hence, *Bmal1* deletion seems to inactivate Wnt/ β -catenin pathway components after peripheral axotomy, at least at 1 dpi.

R2.5. *Fig 6: The authors conclude that “Hence ZT0-3 and ZT12-15 correspond to opposite phases of DRG circadian clock.” (p 14). Opposite phases in circadian biology means the peak and troughs of gene expression are anti-phasic in the context of cyclic expression over 24 hour time. Conclusions about circadian or diurnal expression in this figure, and others, are too strong given that only two time points over 24 hours were tested. First, it is not clear that these represent either the peak or trough of *Bmal1* and *Tet3* expression. Second, it is not clear if these genes are functionally correlated, *Tet3* might be changing due to the light-dark cycle, while *Bmal1* is circadian, and actually both are not opposite to each other at all. Third, sampling only two times can miss rhythms. As shown in 6A the circadian clock gene *Nr1d1* is not even rhythmic, nor are most of the genes tested in Fig S10. When the authors later state: “Having demonstrated diurnal oscillation of *Tet3*-5hmC that is anti-phasic to *Bmal1* transcriptional activity in DRG neurons” (p 16), these statements ignore limitations of the data and are far too strong. This same problem also applies to Fig S11E where the authors claim genes show “transcriptional oscillation” based on only two timepoints tested. If the authors want to conclude rhythms are present, they need to examine many more time points over 24 hours. At least the model proposed in Fig 6E should be tested by examining *Bmal1* protein by antibody at more than one time point to show that it is higher at ZT0-3.*

>> We agree with the Reviewer the need for detailed mapping of time-of-day effect. We have conducted new IF studies at 4-hour intervals along ZT points, and this revealed zenith of *Tet3* and 5hmC expression in DRG neurons at ZT12, and nadir of *Tet3* expression at ZT0 and 5hmC at ZT4 (new **Fig. S14a**). Interestingly, *Bmal1* protein and mRNA oscillations in DRG are not entirely in sync, echoing reported lag time between *Bmal1* mRNA and protein levels^{16, 17}. In this context, *it is noteworthy that *Bmal1* protein levels do not directly equate *Bmal1* DNA binding or transcriptional activity*, which is also controlled by regulators of the feedback loops such as *Per* and *Cry*¹⁸. Indeed, a recent study reported *Bmal1* level peaking at the onset of dark phase¹⁹, which paradoxically coincided with its dissociation from DNA and repression by *Per/Cry* complex^{16, 17, 20}. We therefore simplified the model (now shown in **Fig. 8e**).

We also conducted neurite outgrowth assays on primary DRG neurons isolated every 6 hrs along ZT period and found that DRG neurons isolated at ZT12 extended the longest neurites while those isolated at ZT0 extended the shortest neurites (new **Fig. S16a**).

As mentioned above, we agree with the Reviewer that future studies are needed to disentangle circadian (mediated by *Bmal1* clock) vs. light entrainment (affecting sleep and neuronal activity) that controls Tet3-5hmC rhythms (using mice kept in constant darkness or light shift experiments). We also revised our manuscript to tone down circadian control, and omitted the statement that Tet3 and *Bmal1* are opposite of each other in diurnal changes.

*R2.6. Fig 8: The authors show that the *Bmal1* cKO has the same daily change in neurite growth and regeneration as the control. This data suggests *Bmal1* regulates *Tet3* non-rhythmically, acting as a negative regulator but not necessarily in a circadian manner. The data showing that *Bmal1* decreases during injury (Fig 2C-D), further suggests that *Bmal1* cannot be responsible for the time of day changes in neuron regeneration – it is absent to play a role during the stress response. These data challenge the schematic shown in Fig 8H. Another factor must be responsible, maybe light-response, sleep, or other neuronal activity that coordinate with *Bmal1* to effect these changes. The authors should consider these. Alternatively, it is possible the authors have missed the circadian interaction by only testing at two times of day. What are *Tet3* expression and 5hmC levels in the *Bmal1* cKO at these and different times? Sampling at only the morning and evening may have missed a stronger interaction between *Bmal1* and *Tet3*/regeneration, a maximum or minimum that is not ZT0-3 or ZT12-15 that would better support the model the authors propose.*

>> We appreciate Reviewer's comment and have followed the advice to include alternative models that *light entrainment, neuronal activity, or sleep* may underlie the diurnal oscillation observed here. Of note, our studies were conducted with *Bmal1*^{cKO} mice maintained in regular light:dark cycle, thus the continued time-of-day effect observed in DRGs from *Bmal1*^{cKO} mice likely reflect “masking”, a response to light overriding the absence of circadian clock⁸⁻¹⁰. To distinguish circadian vs. light entrainment, future studies are needed by keeping *Bmal1*^{cKO} mice in constant dark cycle or conduct light shift assays (see discussion above).

Regarding *Bmal1* downregulation (*not completely absent*) after peripheral axotomy, we would like to clarify that our data showed that *Bmal1* downregulation was more pronounced for PL at ZT12-15 than at ZT0-3, thus *Bmal1* may still exert an influence in gating regenerative responses after injury. Moreover, at morning vs. evening, we found Tet3/5hmC epigenetic differences, thus DRG might be primed differently in response to injury, with different sets of RAGs induced when injury occurring at different ZT (please see our new analysis

in new **Fig. S17**, and our response to Reviewer 3 (R3.6)). Certainly, light cycle, sleep, and neuronal activity may also be determining factors.

Following the advice, we have added new studies with neurite outgrowth assays on DRGs neurons from *Bmal1^{cKO}* mice (kept in regular light:dark cycle), sampling every 6 hrs. Of note, since the process of dissociation and plating primary DRG neurons takes more than 3 hrs, this assay only provides a rough estimate of axon regrowth potential at a particular ZT. Results showed longer neurites when DRG neurons were collected at ZT12 (new **Fig. S16b**). Hence, in the absence of *Bmal1*, time-of-day effect from light cycle, sleep, or neuronal activity are still present, consisting with “masking”.

R2.7. Fig S12: The authors show that Creb is higher in the evening (A-B) then show that Creb activation in Bmal1 cKO through cAMP does not further increase neuron growth. Creb should be tested in Bmal1 cKO.

>> We have examined pCREB^{S133} in *Bmal1^{cKO}* DRG neurons, which was higher than in control DRG in response to PL at 3 dpi (see **Fig. 6f**).

R2.8. Statistics: The authors should provide a supplementary table showing full results of the statistics throughout the paper. For example in Fig 8D, a. two-way ANOVA and Bonferroni is done but only the p-value is given.

>> Done. We have now provided a supplementary table listing full results of the statistics throughout the paper.

Reviewer #3:

Halawani et al. describe a new link between the circadian transcriptional regulator BMAL1 and the Tet3 DNA methylcytosine dioxygenase in peripheral axon regeneration. They also report diurnal oscillations in DNA hydroxymethylation that may contribute to differences in axonal growth throughout the day. Together, these are interesting observations, but the key findings are not sufficiently nailed down. This limits unfortunately limited my enthusiasm for the manuscript.

>> We appreciate the positive comment, and have conducted new experiments as follows:

Major:

***R3.1.** A key claim of this paper is that Bmal1 and Tet3 directly interact, but the only data provided to support this is a one-way IP a cell line. I recommend the authors consider IPing with Bmal1 and seeing if Tet3 comes down as well as investigate whether this interaction occurs in DRG.*

>> Done. Please also see our detailed response for Reviewer 1 (R1.2). Briefly, we have performed new co-IP study using DRG lysates, showing that IP with a specific antibody against Tet3¹⁻³ could pull down Tet3 along with Bmal1 in both Neuro-2a and DRG lysates (new **Fig. S2a**). Interestingly, unlike in Neuro-2a lysate with a predominant band of Tet3 at ~185 kDa, we observed a dominant band of Tet3 at ~80 kDa in DRG lysates, while pull down of Tet3 resulted in a faint band at ~185 kDa and multiple bands at lower molecular weights (new **Fig. S2a**), in agreement that Tet3 can be proteolytically processed by Ca²⁺ calpain cleavage, producing ~80 kDa band⁴. Since the Tet3 antibody binds to the C-terminal catalytic domain, these fragments are consistent with N-terminal proteolysis, a susceptible region with low structural complexity (new **Fig. S2b**).

We also attempted reverse co-IP with a specific antibody against the C-terminal region of Bmal1 (validated with Bmal1 knockout cells) but failed to pull down either endogenous or overexpressed Tet3 in Neuro-2a cells (new **Fig. S2c**). This may be related to binding of Tet3 to the C-terminal domain of Bmal1, which contains a conserved transactivation domain prominently involved in binding to transcriptional regulators^{5,6}.

***R3.2.** While the CreER system is very powerful, it is important to confirm the core findings are not secondary to tamoxifen itself. I believe this control is missing for the axon outgrowth, Tet3 expression levels, macrophage activation, etc. in the Bmal1 cKO.*

>> Both cohorts (*Bmal1* cKO and littermate controls) received the same tamoxifen injection.

***R3.3.** The evidence to support the Bmal1 cKO phenotypes are through Tet3 is not strong. There is a very subtle phenotype with the Tet3 inhibitor C35, but this would be stronger with a dose response over wider range as well as some genetic evidence as these tools are widely available.*

>> Following the advice, we have performed dose response study with the Tet inhibitor C35, showing that Tet3 activity is required for the growth promoting effect of *Bmal1* deletion in a dose-dependent manner (new **Fig. S8e**).

Second, to better define Bmal1-controlled genes in regenerating DRG neurons (termed Bmal1 regulon), we have conducted *new RNA-seq studies* on primary DRG neurons from *Bmal1* cKO vs. control mice (new **Figs. 7a-g; Tables S12-S16**). This identified 625 DEGs (*Bmal1*^{cKO} / control), among which 85 genes contain PL specific DhMRs (new **Figs. 7e-f; Table S9-S10**), and 36 genes contain Bmal1 binding motif within the PL-DhMRs (new **Fig. 7g; Tables S11**). Therefore, we agree with the Reviewer that Bmal1 regulates axon regenerative gene program through Tet3 dependent and independent mechanisms. We have thus toned down the link of Bmal1 and Tet3, and clarified that the axon promoting effect of *Bmal1* deletion is partly operated through Tet3-5hmC elevation.

R3.4. Additional time points would be helpful throughout the day would be helpful for understanding the circadian nature of gene expression, DNA hmC and axonal growth.

>> Done. We have conducted mapping of Bmal1 and Tet3-5hmC in DRG neurons every 4 hrs along ZT period, showing zenith and nadir of their expression in DRG neurons (new **Fig. S14a**). We also examined neurite outgrowth of DRG neurons every 6 hrs along ZT period, showing axon growth potential highest at ZT12 and lowest at ZT0 (new **Fig. S16a**). Please also see our responses to Reviewer 2 (R2.5).

R3.5. Bmal1 cKO does not seem sufficient to block the diurnal effects on axonal growth, what factors are driving this? Do Clock, Per KOs display related axonal growth phenotypes after injury?

>> As discussed in response to Reviewer 2, alternative models such as light entrainment, neuronal activity, or sleep, may contribute to diurnal oscillation even in the absence of Bmal1. Indeed, our studies were conducted with *Bmal1*^{cKO} mice maintained in regular light:dark cycle, thus the continued time-of-day effect of DRG in *Bmal1*^{cKO} mice may reflect “masking”, a response to light overriding the absence of circadian clock⁸⁻¹⁰. We have tested that pharmacological stabilization of PER and CRY (negative regulators of BMAL1), which resulted in longer neurite outgrowth in induced neurons (iNeurons) derived from human embryonic stem (hES) cells (**Figs. 3b**). In the current study, we focused on Bmal1, an essential and nonredundant regulator of the mammalian circadian clock⁸. Future studies are worthwhile to test if *Clock* or *Per* KO would display related axon growth phenotypes after injury.

R3.6. Are there diurnal changes in RAG induction and 5hmC that are affected by Bmal1 cKO? 5hmC

immunofluorescence is rather non-specific. Locus specific measurements would be more compelling and mechanistically interpretable.

>> As another indicator of the time-of-day effect on RAG induction, we compared the DEGs profiled in axotomized DRG with PL performed at ZT2-4 (this study) and with published RNA-seq with PL at unspecified ZT time ^{21, 22} (new **Fig. S17a**). This revealed 815 shared genes featuring Circadian Rhythm, Axonal guidance, cAMP-mediated pathway, CREB, HIF, and Xenobiotic metabolism, as well as immune signaling (new **Figs. S17b-c; Table S13**). But notably, there was a large fraction of non-overlapping DEGs: 73% of the DEGs from this study and 58% of the published study ^{21, 22}. Hence, RAGs induction is affected by time-of-day, with shared but also unique genes. Future locus-specific measurement of 5hmC reconfigurations at different ZT in response to PL would be needed, but best conducted in purified DRG neurons (technically more involved).

Minor:

R3.7. Is the difference in Fig 1F statistically significant? Would increased ATAC signal be seen at any gene that is unregulated after PL regardless of its hydroxymethylation status?

>> We added statistical information on the differences of the ATAC signals at DhMRs and TSS of the associated genes (see **Fig. 1h**, new **S1d**).

We have also separately analyzed PL-DhMRs with 5hmC gain or loss (following the advice from Reviewer 1), and found that the increased ATAC signals at DhMRs after PL were more pronounced for the group showing 5hmC gain than loss (new **Fig. S1d**). DNA hydroxymethylation is just one epigenetic mechanism, it is likely that many RAGs induced by PL would show increased chromatin access; the ATAC data provided just one piece of information of the chromatin state of the PL-DhMRs and associated genes.

R3.8. How specific is the Bmal1 antibody for the IF staining in Fig 2C? This quantification would be more convincing if staining from the Bmal1 f/f cKO were shown.

>> To our knowledge, we used the best characterized anti-Bmal1 antibody in the circadian field (Novus #100-2288), routinely used by the Takahashi and Musiek groups. Additionally, we have validated the antibody in DRGs from *Bmal1* cKO mice (see **Fig. S5e-f**). We have provided more representative images for Fig. 2C (now **Fig. 2e**), showing reduced Bmal1 immunosignals in axotomized DRG after PL.

R3.9. Fig S3E: shouldn't Bmal1 protein only be downregulated in Thyl-positive neurons? These are not displayed in the image, but it appears that all Bmal1 staining is weaker in the example. Co-staining with Cre could help clarify this.

>> Thy-1-Cre is expressed in all DRG neurons (see **Fig. S5a**), consistent with characterization in previous publications²³. Since *Thy1-CRE^{ER}/EYFP* (SLICK-H) transgenic line drives pan-neuronal expression of both CreER recombinase and EYFP, we confirmed uniform expression of EYFP in DRG neurons in both DRG tissue sections and in primary DRG cultures (see **Figs. S5a-b**). IF staining showed that *Bmal1* levels were reduced in all DRG neurons in cKO (now shown in **Fig. S5e**). *Bmal1* was detectable in glial cells in DRGs (at lower levels than in DRG neurons), and it appeared comparable between cKO vs. control (see **Fig. S5f**).

R3.10. Changes in pCREB and b-Catenin levels in Bmal1 cKO mice are not specific to the enriched bioinformatic pathways in Fig 5. It might be more relevant to use inhibitors of various arms of the enriched pathways to see if this affects the downstream up regulation of pCREB or b-catenin.

>> We have conducted new RNA-seq studies with primary DRG neurons from *Bmal1 cKO* vs. control mice (see **Fig. 7a-g**), and results confirmed that CREB and Wnt/ β -catenin pathway components are regulated by *Bmal1*. Future functional studies are needed to interrogate whether these pathways contribute to the promoting effect of *Bmal1 cKO*, as the current study focuses on the link to Tet3-5hmC. We have clarified in the revised manuscript that *Bmal1 cKO* likely operates through Tet3-dependenet and independent mechanisms (e.g., CREB and Wnt/ β -catenin).

R3.11. Recommend copy editing of manuscript as there are a number of grammatical errors and typos

>> We have polished the manuscript to correct typos and grammatical errors.

References

1. Han, L., *et al.* Embryonic defects induced by maternal obesity in mice derive from Stella insufficiency in oocytes. *Nat Genet* **50**, 432-442 (2018).
2. Bauer, C., *et al.* Phosphorylation of TET proteins is regulated via O-GlcNAcylation by the O-linked N-acetylglucosamine transferase (OGT). *J Biol Chem* **290**, 4801-4812 (2015).
3. Chamesian, A.G., *et al.* 5-Hydroxymethylcytosine (5hmC) and Ten-eleven translocation 1-3 (TET1-3) proteins in the dorsal root ganglia of mouse: Expression and dynamic regulation in neuropathic pain. *Somatosens Mot Res* **34**, 72-79 (2017).
4. Wang, Y. & Zhang, Y. Regulation of TET protein stability by calpains. *Cell Rep* **6**, 278-284 (2014).
5. Gustafson, C.L., *et al.* A Slow Conformational Switch in the BMAL1 Transactivation Domain Modulates Circadian Rhythms. *Mol Cell* **66**, 447-457.e447 (2017).
6. Xu, H., *et al.* Cryptochrome 1 regulates the circadian clock through dynamic interactions with the BMAL1 C terminus. *Nat Struct Mol Biol* **22**, 476-484 (2015).
7. Renthall, W., *et al.* Transcriptional Reprogramming of Distinct Peripheral Sensory Neuron Subtypes after Axonal Injury. *Neuron* **108**, 128-144 e129 (2020).
8. Bunger, M.K., *et al.* Mop3 is an essential component of the master circadian pacemaker in mammals. *Cell* **103**, 1009-1017 (2000).
9. Musiek, E.S., *et al.* Circadian clock proteins regulate neuronal redox homeostasis and neurodegeneration. *J Clin Invest* **123**, 5389-5400 (2013).
10. Mieda, M. & Sakurai, T. *Bmal1* in the nervous system is essential for normal adaptation of circadian locomotor activity and food intake to periodic feeding. *J Neurosci* **31**, 15391-15396 (2011).

11. Stephan, F.K. & Zucker, I. Circadian rhythms in drinking behavior and locomotor activity of rats are eliminated by hypothalamic lesions. *Proc Natl Acad Sci U S A* **69**, 1583-1586 (1972).
12. Kwon, I., *et al.* BMAL1 shuttling controls transactivation and degradation of the CLOCK/BMAL1 heterodimer. *Mol Cell Biol* **26**, 7318-7330 (2006).
13. Lipton, J.O., *et al.* The Circadian Protein BMAL1 Regulates Translation in Response to S6K1-Mediated Phosphorylation. *Cell* **161**, 1138-1151 (2015).
14. Patel, A.K., Park, K.K. & Hackam, A.S. Wnt signaling promotes axonal regeneration following optic nerve injury in the mouse. *Neuroscience* **343**, 372-383 (2017).
15. Poy, F., Lepourcelet, M., Shivdasani, R.A. & Eck, M.J. Structure of a human Tcf4-beta-catenin complex. *Nat Struct Biol* **8**, 1053-1057 (2001).
16. Lee, C., Etchegaray, J.P., Cagampang, F.R., Loudon, A.S. & Reppert, S.M. Posttranslational mechanisms regulate the mammalian circadian clock. *Cell* **107**, 855-867 (2001).
17. Tamaru, T., *et al.* Nucleocytoplasmic shuttling and phosphorylation of BMAL1 are regulated by circadian clock in cultured fibroblasts. *Genes Cells* **8**, 973-983 (2003).
18. Buhr, E.D. & Takahashi, J.S. Molecular components of the Mammalian circadian clock. *Handb Exp Pharmacol*, 3-27 (2013).
19. Welz, P.S., *et al.* BMAL1-Driven Tissue Clocks Respond Independently to Light to Maintain Homeostasis. *Cell* **177**, 1436-1447.e1412 (2019).
20. Menet, J.S., Pescatore, S. & Rosbash, M. CLOCK:BMAL1 is a pioneer-like transcription factor. *Genes Dev* **28**, 8-13 (2014).
21. Hervera, A., *et al.* Reactive oxygen species regulate axonal regeneration through the release of exosomal NADPH oxidase 2 complexes into injured axons. *Nat Cell Biol* **20**, 307-319 (2018).
22. Palmisano, I., *et al.* Epigenomic signatures underpin the axonal regenerative ability of dorsal root ganglia sensory neurons. *Nat Neurosci* **22**, 1913-1924 (2019).
23. Young, P., *et al.* Single-neuron labeling with inducible Cre-mediated knockout in transgenic mice. *Nat Neurosci* **11**, 721-728 (2008).

REVIEWER COMMENTS

Reviewer #1 (Remarks to the Author):

The authors have addressed each of the reviewer's concerns in a detailed manner, providing additional data or modifications where necessary. These changes have further strengthened the paper. I would like to recommend this paper for publication.

Reviewer #2 (Remarks to the Author):

The authors have addressed my comments. Although a direct relationship consisting of Bmal1 regulating a circadian rhythm in neuronal repair does not seem to be the only factor at play, the data presented are solid and of interest.

Reviewer #3 (Remarks to the Author):

The revised manuscript by Halawani et al. addresses many but not all of the reviewer concerns. A few important questions remain:

1. The link between Tet3 and Bmal1 does not seem nailed down. Without a clear co-IP, I'd suggest the authors present a more convincing 1-way IP with KO controls.
2. The hypothesis that Tet3 plays any role in Bmal1's effect on axonal regeneration seems derived from correlative gene expression overlaps and an inhibitor which could have off-target effects. While I am pleased that the authors have toned down the discussion to indicate there are likely Tet3-independent mechanisms, I'm not sure there is sufficient data to say there are definitely Tet3-dependent mechanisms at all. Are we sure the Tet3 effect isn't an entirely independent pathway from Bmal1?
3. The authors have done a much better job describing the diurnal changes in neurite outgrowth in Fig S16a. Given the variability that may be seen from culture-to-culture (even at the same ZTA), please comment on how many independent cultures each time point is derived.

RESPONSE LETTER

Point-by-point response:

Reviewer #1:

The authors have addressed each of the reviewer's concerns in a detailed manner, providing additional data or modifications where necessary. These changes have further strengthened the paper. I would like to recommend this paper for publication.

>> We appreciate the positive evaluation.

Reviewer #2:

*The authors have addressed my comments. Although a direct relationship consisting of *Bmal1* regulating a circadian rhythm in neuronal repair does not seem to be the only factor at play, the data presented are solid and of interest.*

>> We appreciate the positive evaluation. We further clarified in Discussion that *Bmal1*-circadian rhythm is only one factor at play for neural repair after injury.

Reviewer #3:

The revised manuscript by Halawani et al. addresses many but not all of the reviewer concerns. A few important questions remain:

*1. The link between *Tet3* and *Bmal1* does not seem nailed down. Without a clear co-IP, I'd suggest the authors present a more convincing 1-way IP with KO controls.*

>> First, to further evaluate a functional link of *Tet3* and *Bmal1*, we conducted a new epistasis study (see below in answer to comment #2), which demonstrated that *Tet3* knockdown by siRNA also reduced the growth advantage of *Bmal1*-deficient DRG neurons (**new Fig. 5e-f**). Thus, this new genetic evidence complements the pharmacological approach with *Tet3* inhibitor C35 in showing the contribution of *Tet3* for the growth phenotype of *Bmal1* cKO DRG neurons; but this does not preclude involvement of other factors or *Tet3*-independent mechanisms, a point we further clarified in the Discussion.

Second, following the advice, we conducted 1-way IP with negative controls. As mentioned in our manuscript, Western blot could not detect a reduction of *Bmal1* protein levels in whole DRG tissue from *Bmal1* cKO mice (likely due to abundant glial cells in DRG), thus DRGs from *Tet3* or *Bmal1* full KO would be ideal, but unfortunately we do not have available these mutant mice. We therefore performed co-IPs of endogenous *Tet3* and *Bmal1* from Neuro-2a cells after *Tet3* knockdown (si*Tet3* vs. siCTL) by pulling down *Tet3* and immunoblotting for *Bmal1* (we kindly remind the Reviewer that the available *Bmal1* antibodies could not co-IP *Tet3*, thus reverse co-IP was not performed at this time).

For the *Tet3* siRNA knockdown study, due to the constraint of siRNA transfection reagents required, we had to significantly reduce the cell numbers. In our original co-IP studies (shown in Fig. 1F, now in **Fig. S2a**, also included in **Figure A(a)** below for comparison), we used 8.5×10^6 Neuro-2a cells cultured in a T75 flask and collected the lysates 4 days later with cells reaching high confluence. For the *Tet3* siRNA study, we plated 3.5×10^5 cells in each well of the 6-well plate and collected lysates 72 hrs after siRNA transfection. Under these conditions,

as shown in **Figure A(b)**, co-IP with anti-Tet3 pulldown with short exposure of WB membrane showed hardly detectable Bmal1. Longer exposure seemed to show a reduction of Bmal1 co-IP with *Tet3* knockdown (44% as compared to siCTL), although the result was suboptimal due to background noise. We suspect that the requirement for a large

number of cells as starting material for successful co-IP may reflect a *transient nature of Tet3-Bmal1 interaction, poor Tet3 antibody affinity, or low abundance of endogenous Tet3.*

Given these limitations, we moved the co-IP data from Fig. 1F to supplemental **Fig. S2a** and revised the manuscript by highlighting the functional link of Bmal1 and Tet3. A direct Bmal1-Tet3 interaction might be relevant but is *not the main focus of the study*, especially since we showed that *Bmal1* deletion and circadian oscillation affect Tet3 mRNA level (Fig. 5C, Fig. S13b and Fig. 8c), thus pointing to a transcriptional control mechanism perhaps through REV-ERBs (Fig. S5d). We added in Discussion that a detailed analysis of a possible protein-protein interaction is worthwhile for future investigation.

2. The hypothesis that Tet3 plays any role in Bmal1's effect on axonal regeneration seems derived from correlative gene expression overlaps and an inhibitor which could have off-target effects. While I am pleased that the authors have toned down the discussion to indicate there are likely Tet3-independent mechanisms, I'm not sure there is sufficient data to say there are definitely Tet3-dependent mechanisms at all. Are we sure the Tet3 effect isn't an entirely independent pathway from Bmal1?

>> We have conducted a new epistasis study with *Tet3* knockdown in *Bmal1* cKO DRG neurons as compared to controls (**new Fig. 5e-f**). Echoing the results from the pharmacological approach, we found that *Tet3* siRNA

knockdown significantly reduced the growth promoting effect of *Bmal1* deletion in neurite outgrowth assay. Thus, genetic evidence complements our pharmacological results in supporting a functional link between Tet3 and *Bmal1* in controlling neurite outgrowth.

Aside from the epistatic data showing that the growth advantage of *Bmal1*^{CKO} DRG neurons requires Tet3, other findings from our study also supports a transcriptional link of *Bmal1* and Tet3: i) *Bmal1* cKO led to Tet3 upregulation and global 5hmC gains in DRG neurons, ii) time-of-day changes of Tet3 and 5hmC levels in DRG neurons correspond to *Bmal1* oscillation, iii) *Bmal1* binding motif is enriched in DhMRs (regulated by Tet3 enzyme) in response to peripheral lesion. Altogether, these results support a transcriptional and functional link of *Bmal1* and Tet3 in DRG neurons.

3. The authors have done a much better job describing the diurnal changes in neurite outgrowth in Fig S16a. Given the variability that may be seen from culture-to-culture (even at the same ZTA), please comment on from how many independent cultures each time point is derived.

>> We clarified in **Fig. S16a** legend that for each ZT time point, the data were collected from 300 DRG neurons from n=2-3 mice per condition.

Neurite outgrowth assays on primary DRG neurons typically show a large variability of neurite length (e.g., Fig. 3e, Fig. 5e-f, and Fig. 5h), which may reflect heterogenous populations of DRG neurons and variations in how close the axotomy site is to the soma during dissociation. Further contributing to the variability is that each cultured DRG neuron extends multiple axonal branches, thus the experimental standard of the field is to quantify the length of the longest axon. Our approach to reduce variability is to always collect the DRGs from the same levels (L4-L6) and to quantify a large number of neurons from multiple mice for each condition.

REVIEWERS' COMMENTS

Reviewer #3 (Remarks to the Author):

The authors have addressed my primary concerns either with new data or text clarification.

RESPONSE LETTER

Point-by-point response:

REVIEWERS' COMMENTS

Reviewer #3 (Remarks to the Author):

The authors have addressed my primary concerns either with new data or text clarification.

> We appreciate the positive response from Reviewer 3.